# Med23 serves as a gatekeeper of the myeloid potential of hematopoietic stem cells

Xufeng Chen [1], Jingyao Zhao[1], Chan Gu[2], Yu Cui[1], Yuling Dai[1], Guangrong Song[3], Haifeng Liu[1], Hao Shen[1], Yuanhua Liu [4], Yuya Wang[1], Huayue Xing[1], Xiaoyan Zhu[1], Pei Hao [4], Fan Guo [2] & Xiaolong Liu [1,3]

In response to myeloablative stresses, HSCs are rapidly activated to replenish myeloid progenitors, while maintaining full potential of self-renewal to ensure life-long hematopoiesis. However, the key factors that orchestrate HSC activities during physiological stresses remain largely unknown. Here we report that Med23 controls the myeloid potential of activated HSCs. Ablation of Med23 in hematopoietic system leads to lymphocytopenia. Med23-deficient HSCs undergo myeloid-biased differentiation and lose the self-renewal capacity. Interestingly, Med23-deficient HSCs are much easier to be activated in response to physiological stresses. Mechanistically, Med23 plays essential roles in maintaining stemness genes expression and suppressing myeloid lineage genes expression. Med23 is downregulated in HSCs and Med23 deletion results in better survival under myeloablative stress. Altogether, our findings identify Med23 as a gatekeeper of myeloid potential of HSCs, thus providing unique insights into the relationship among Med23-mediated transcriptional regulations, the myeloid potential of HSCs and HSC activation upon stresses.

[1] State Key Laboratory of Cell Biology, CAS Center for Excellence in Molecular Cell Science, Institute of Biochemistry and Cell Biology, University of Chinese Academy of Sciences, Chinese Academy of Sciences, 200031 Shanghai, China. [2] Center for Translational Medicine, Ministry of Education Key Laboratory of Birth Defects and Related Diseases of Women and Children, Department of Obstetrics and Gynecology, West China Second University Hospital, Sichuan University, 610041 Chengdu, Sichuan, China. [3] School of Life Science and Technology, ShanghaiTech University, 200031 Shanghai, China. [4] Key Laboratory of Molecular Virology and Immunology, Institut Pasteur of Shanghai, Chinese Academy of Sciences, 200031 Shanghai, China. These authors contributed equally: Xufeng Chen, Jingyao Zhao. Correspondence and requests for materials should be addressed to F.G. (email: guofan@scu.edu.cn) or to X.L. (email: liux@sibs.ac.cn)

Hematopoietic stem cells (HSCs), which harbor the capacities of both self-renewal and differentiation, are responsible for the life-long production of all blood cells[1]. The life-long maintenance of hematopoiesis is a highly orchestrated process that requires the balanced differentiation output of the myeloid and lymphoid lineages as well as the precise regulation of the size of the HSC pool through HSC self-renewal, which is based on hematopoietic demands and the needs of the organism[2]. Thus, the self-renewal capacity and the balanced differentiation potential of HSCs are essential for sustaining the homeostasis of the entire hematopoietic system in both physiological and pathological conditions[3].

HSCs resident in adult bone marrow (BM) are principally quiescent in steady state. However, upon physiological stresses (e.g., bleeding, irradiation, transplantation, myelosuppressive chemotherapy etc.), HSCs are quickly activated and generate workhorse progenitors in order to replenish their tissue constituents. So far, both intrinsic and extrinsic factors have been revealed to regulate the activation of HSCs. The extrinsic cell signals, such as IFNg and IFNa can efficiently activate the quiescent HSCs[4,5]. Moreover, the cell intrinsic factors CDK6 and Egr1 control the activation of HSCs through cell-cycle regulation and transcriptional regulation respectively[6,7]. However, little is known about how HSC activities are orchestrated during psychological stresses and the key factors that control this process remain largely unknown.

The mammalian mediator complex is an evolutionarily conserved multiprotein complex that bridges between gene-specific transactivators and the RNA polymerase II-associated basal transcription machinery[8,9]. As a key subunit of the transcriptional mediator complex, Med23 (also known as Sur2) plays an important role in orchestrating the transcriptional profiles in various cell types[10–15]. It has been reported that Med23 regulates the mitogen-activated protein kinase signaling pathway through mediating the responses of several immediate early genes (IEG) to serum mitogens[16,17]. Moreover, Med23 has important roles in diverse biological processes including adipogenesis, brain development, cell differentiation and carcinogenesis[12,14,18,19]. Our findings have suggested that Med23 controls the activation threshold of T-cell at the transcriptional level, and therefore prevents the development of autoimmunity[15].

To investigate the functions of Med23 in HSCs, we conditionally deleted Med23 in the hematopoietic system. Med23 deficiency leads to lymphocytopenia but normal myeloid outcomes, and loss of renewal capacity. Med23-deficient HSCs are more activated with enhanced myeloid potential, which, in turn, contributes to a better survival after serial 5-FU administration. Mechanistically, Med23 prevents HSCs from the myeloid potential by suppressing the expression of the myeloid-specific genes and maintaining the expression of HSC stemness genes. Altogether, our findings identify Med23 as a key factor that controls the myeloid potential of HSCs by repressing the expression of myeloid-related genes, thus providing unique insights into the relationship among Med23-mediated transcriptional regulations, the myeloid potential of HSCs and HSC activation upon stresses.

## Results

**Med23 deficiency leads to lymphocytopenia.** As most of the mediator subunits, Med23 was ubiquitously expressed in different hematopoietic cells (Supplementary Fig. 1a). To study the function of Med23 in hematopoiesis, we bred conditional $Med23^{L/L}$ mice with the inducible *Mx1-cre* strain to ablate Med23 in adult hematopoietic system[15,20]. *Med23* was efficiently ablated from the

**Table 1 Complete blood counts of peripheral blood in WT and KO mice**

| Parameter | WT | KO | *p*-value |
|---|---|---|---|
| White blood cells ($10^9$/L) | 11.0 ± 1.3 | 5.7 ± 0.3 | 0.0086 |
| Lymphocytes ($10^9$/L) | 8.5 ± 0.6 | 4.2 ± 0.8 | 0.0033 |
| Neutrophils ($10^9$/L) | 2.1 ± 0.5 | 1.2 ± 0.1 | 0.0731 |
| Monocytes ($10^9$/L) | 0.14 ± 0.03 | 0.10 ± 0.04 | 0.3164 |
| Red blood cells ($10^{12}$/L) | 10.0 ± 0.02 | 10.7 ± 0.09 | 0.0005 |
| Hemoglobin (g/L) | 147.3 ± 2.4 | 140.7 ± 3.3 | 0.0807 |
| Platelets ($10^{12}$/L) | 1.41 ± 0.18 | 0.827 ± 0.11 | 0.0161 |

Peripheral blood was collected from WT and KO mice at 21 days post poly(I:C) administration and analyzed for complete blood count ($n = 3$). Data are shown as mean ± SD

hematopoietic system after poly(I:C) administration and loss of Med23 did not affect the stability of the entire mediator complex (Supplementary Fig. 1b, c, e). Compared with $Mx1\text{-}cre^-Med23^{L/L}$ control littermates (designated WT mice), $Mx1\text{-}cre^+Med23^{L/L}$ mice (designated KO mice) showed decreased number of lymphocytes in peripheral blood (Table 1) and decreased cellularity in BM, spleen (Spl), and thymus (Thy) compared with WT controls (Fig. 1a). Interestingly, both T-cell subsets in different developmental stages in Med23-deficient Thy and the mature T cells in the Med23-deficient Spls were significantly decreased compared with those from WT controls (Supplementary Fig. 3a and Fig. 1b). Besides, decreased cellularity in B cell progenitors could be seen in Med23-deficient BMs, leading to the reduction of mature B cells in both the BMs and the Spls from $Med23^{-/-}$ mice compared with those from WT mice (Fig. 1c and Supplementary Fig. 3b). Moreover, the NK cells were also significantly decreased in Med23-deficient BMs and Spls (Fig. 1d). However, in spite of the slightly changes of the frequency of myeloid cells, the total number of different myeloid cells in both BMs and Spls remained unchanged after Med23 ablation (Fig. 1e and Supplementary Fig. 3c). These findings suggested that Med23 deficiency led to lymphocytopenia.

**Loss of Med23 results in increased myeloid progenitors.** Given that Med23 deficiency led to lymphocytopenia but did not affect myeloid cell number, we compared the myeloid and lymphoid outcomes using the ratio of pooled myeloid cells to pooled lymphoid cells in the BMs and Spls. As expected, the myeloid/lymphoid ratio showed a two-fold increase in Med23-deficient BMs and Spls compared with the WT controls (Fig. 2a), suggesting a development-bias in Med23-deficient mice. This observation prompted us to evaluate the myeloid progenitors, as well as the lymphoid progenitors in Med23-deficient mice. Compared with the WT controls, all the myeloid progenitors including the common myeloid progenitors (CMPs, $CD34^+CD16/32^-$ LK), the granulocyte-macrophage progenitors (GMPs, $CD34^+CD16/32^+$ LK), the megakaryocyte progenitors (MkPs, $CD150^+CD41^+$ LK), as well as the megakaryocyte/erythroid progenitors (MEPs, $CD34^-CD16/32^-$ LK) were significantly increased in the absence of Med23 (Fig. 2b, c and Supplementary Fig. 2a and 3d). Contrastively, consistent with the lymphocytopenia seen in Med23-deficient mice, the common lymphoid progenitors (CLP, $Lineage^-cKit^{low}Sca1^{low}IL7Ra^+$) were significantly decreased in Med23-deficient BMs compared with WT controls (Fig. 2d, e). In addition, Med23-deficient mice showed strongly reduced MPP4 population ($CD34^+CD135^+CD150^-CD48^+LSK$, a subset with significant lymphoid-biased potential in LSK population[21], Supplementary Fig. 3e). These data highlighted the possibility that Med23-deficient HSCs had enhanced myeloid potential but impaired lymphoid potential.

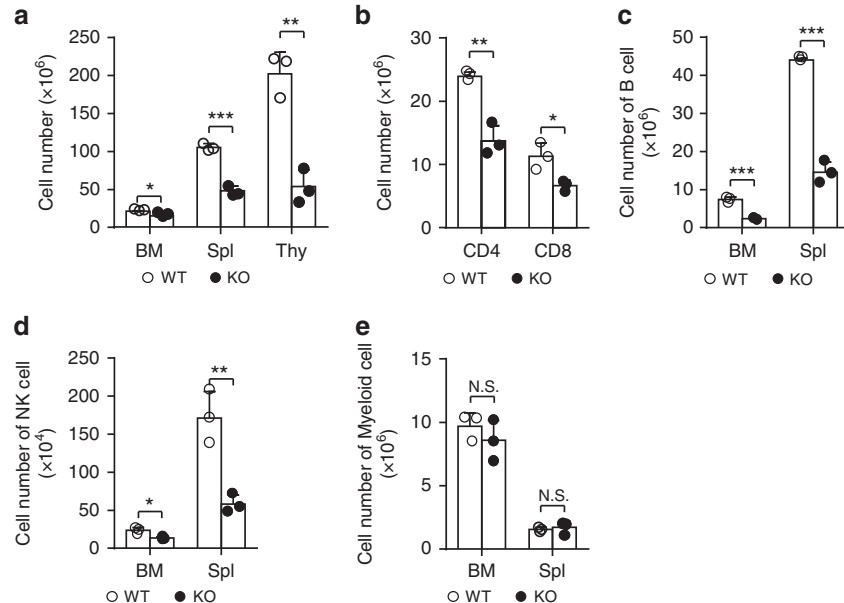

**Fig. 1** Lymphocytopenia in med23-deficient mice. **a** Absolute cell numbers of bone marrow (BM), spleen (Spl), and thymus (Thy) from WT and KO mice ($n = 3$). **b** Absolute cell numbers of CD4 (CD4$^+$CD8$^-$) and CD8 (CD4$^-$CD8$^+$) T cells in spleens of WT and KO mice ($n = 3$). **c** Absolute cell numbers of B cells (B220$^+$) in bone marrows and spleens of WT and KO mice ($n = 3$). **d** Absolute cell numbers of NK cells (NK1.1$^+$) in bone marrows and spleens of WT and KO mice ($n = 3$). **e** Absolute cell numbers of myeloid cells (Gr1$^+$/Mac1$^+$) in bone marrows and spleens of WT and KO mice ($n = 3$). The data are means ± S.D., for all panels: *$p < 0.05$, **$p < 0.01$, ***$p < 0.001$ by Student's $t$-test, N.S. no significance

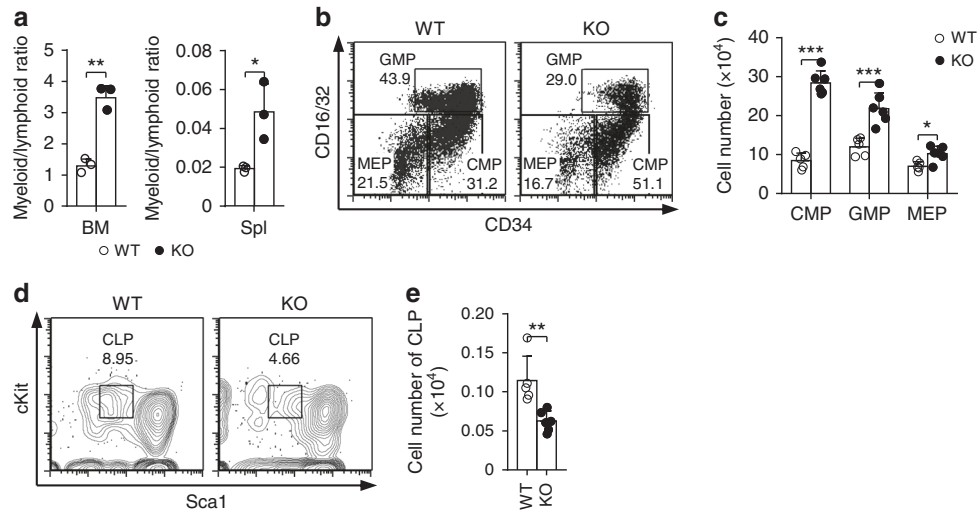

**Fig. 2** Loss of Med23 results in increased myeloid progenitors. **a** The ratio of myeloid cells to lymphoid cells in bone marrows (left) and spleens (right) of WT and KO mice ($n = 3$). **b, c** Representative dot plots Lin$^-$CD127$^-$cKit$^+$Sca1$^-$ live cells (**b**) and cellularity (**c**) of common myeloid progenitors (CMPs), granulocyte-macrophage progenitors (GMPs) and megakaryocyte/erythroid progenitors (MEPs) in bone marrows of WT and KO mice (WT, $n = 5$; KO, $n = 6$). **d, e** Representative contour plots gating from Lin$^-$CD127$^+$ live cells (**d**) and cellularity (**e**) of CLPs in bone marrows of WT and KO mice (WT, $n = 5$; KO, $n = 6$). The data are means ± S.D., for all panels: *$p < 0.05$, **$p < 0.01$, ***$p < 0.001$ by Student's $t$-test

Notably, the increased myeloid progenitors didn't result in the expansion of the mature myeloid cells, suggesting that Med23 might also play a role in the developmental processes from myeloid progenitors to mature myeloid cells.

**Med23-deficient HSCs harbor enhanced myeloid potential.** Given that HSCs give rise to both myeloid progenitors and lymphoid progenitors, we first examined the HSC compartments in Med23-deficient mice. In spite of the somewhat slightly changed expression of stem cell markers (Supplementary Fig. 2b and 4a), long-term HSCs (LT-HSCs, CD34$^-$CD135$^-$ Lineage$^-$cKit$^+$Sca1$^+$, CD34$^-$CD135$^-$ LSK) were increased in Med23-deficient BMs compared with WT control (Fig. 3a, b). By using another set of markers (SLAM markers) to characterize HSCs, we confirmed the increase of HSCs (CD150$^+$CD48$^-$ LSK) in Med23-deficient BMs (Fig. 3c, d and Supplementary Fig. 4b). Interestingly, the increased number of LT-HSCs in Med23-deficient BMs was not due to better survival (Supplementary Fig. 4c, d) or faster cell cycle (Supplementary Fig. 4e).

The increased number of myeloid progenitor cells prompted us to study the possibility that HSCs might have enhanced myeloid potential upon Med23 deletion. To validate this possibility, we

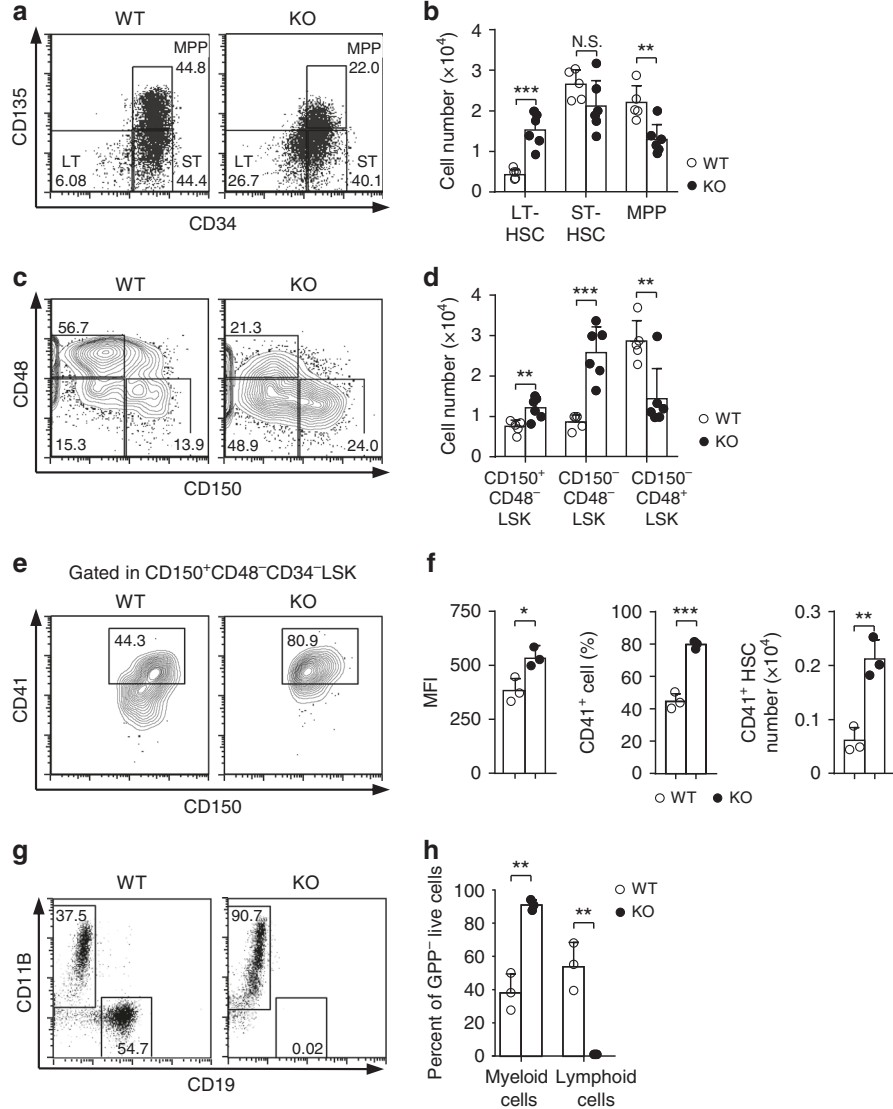

**Fig. 3** *Med23*-deficient HSCs harbor enhanced myeloid potential. **a**, **b** Representative dot plots gating from LSK (**a**) and absolute cell numbers (**b**) of LT-HSC (CD34−CD135− LSK), ST-HSC (CD34+CD135− LSK) and MPP (CD34+CD135+ LSK) from WT and KO mice (WT, n = 5; KO, n = 6). **c**, **d** Representative contour plots gating from LSK (**c**) and absolute cell numbers (**d**) of HSC (CD150+CD48− LSK), CD150−CD48− LSK, and CD150−CD48+ LSK from WT and KO mice using SLAM code (WT, n = 5; KO, n = 6). **e** Representative contour plots showing CD41 expression within HSCs in WT and KO mice. **f** Mean fluorescence intensity (MFI) of CD41 expression in HSCs (left), percent of CD41+ population (middle) within HSCs, and cellularity of CD41+ HSCs (right) in the bone marrows of WT and KO mice (n = 3). **g** Representative FACS plots gating from live cells of WT HSCs and Med23-deficient HSCs co-cultured with OP9 cell line on day 12. **h** Percent of myeloid cells (CD11b+) and lymphoid cells (CD19+) differentiated from WT HSC and Med23-deficient HSC co-cultured with OP9 cell line (n = 3). The data are means ± S.D., for all panels: *p < 0.05, **p < 0.01, ***p < 0.001 by Student's t-test, N.S. no significance

checked the expression of CD41, which marks HSCs with strong myeloid differentiation potential[22]. Compared with WT HSCs, the proportion of HSCs that highly expressed CD41 was significantly higher in *Med23*-deficient HSCs (Fig. 3e, f), suggesting that HSCs harbor enhanced myeloid potential in the absence of Med23. To further compare the myeloid versus lymphoid potential of Med23-deficient HSCs, in vitro differentiation assays of Med23-deficient HSCs were performed using with the OP9 coculture system, which is reported to induce HSCs into both myeloid and lymphoid lineages[23]. As expected, Med23-deficient HSCs underwent myeloid-biased differentiation with impaired lymphoid outputs compared with WT HSCs (Fig. 3g, h). Collectively, these findings implied that Med23 was required for the balance of lymphoid versus myeloid differentiation of HSCs.

Since poly(I:C) treatment may induce stress and activate HSCs[24], the increased HSC number and myeloid-bias potential seen in Med23-deficient HSCs might result from the physiological stress induced by poly(I:C). To confirm the phenotype, tamoxifen-induced Cre system (UBC-cre/ERT2) was used to delete Med23. Upon tamoxifen treatment, Med23 was efficiently deleted (Supplementary Fig. 1d). Notably, the reduced expression of cKit and increased cell number of LT-HSC (CD34−CD135− LSK) were seen in UBC-cre/ERT2;*Med23*fl/fl mice (Supplementary Fig. 5a−c). Moreover, the proportion and absolute cell number of myeloid progenitors (CMP, GMP, MEP, and MkP) were increased, while the lymphoid progenitors (CLP) were decreased in the BM of UBC-cre/ERT2;*Med23*fl/fl mice (Supplementary Fig. 5d, e). More importantly, as seen in Mx1-Cre;*Med23*fl/fl mice, the proportion of HSCs that highly

expressed CD41 was significantly higher in UBC-cre/ERT2;*Med23*fl/fl mice, compared with WT HSCs (Supplementary Fig. 5f,g). These data suggested that HSCs from UBC-cre/ERT2;*Med23*fl/fl mice phenocopied the HSCs from Mx1-Cre;*Med23*fl/fl mice and the poly (I:C) treatment had minimal effects on Med23-deficient phenotypes.

**Med23 deficiency impairs the self-renewal capacity of HSCs.** To test whether Med23 is physiologically required for HSC self-renewal, competitive BM transplantations were carried out to analyze the self-renewal capacity of *Med23*-deficient HSCs (Supplementary Fig. 6a). *Med23*-deficient HSCs failed to reconstitute the irradiated mice in 1st transplantations (Fig. 4a). Moreover, *Med23*-deficient HSCs could not reconstitute the HSC pool in 1st recipient mice and therefore showed no reconstitution in 2nd recipient mice (Fig. 4b–d). These data suggest that Med23 deletion produces a cell-autonomous self-renewal defect in HSCs. Then whole BM transplantations were performed to exclude the possibility that Med23 deficiency might lead to marker changes in HSCs (Supplementary Fig. 6b). Consistently, whole BM cells from *Med23*-deficient mice reconstituted the recipient mice at a much lower level compared with WT BM cells in 1st transplantations (Fig. 4e). Moreover, *Med23*-deficient HSCs could not reconstitute the HSC pool after transplantation, as they could not reconstitute the 2nd recipient mice, leading to lethality (Fig. 4f, g). To substantiate these data, *Mx1-cre⁻Med23*L/L or *Mx1-cre⁺Med23*L/L BM cells were first transplanted without inducing Med23 deletion. The recipient mice were then injected with poly(I:C) at 4-week post-transplantation to induce Med23 deletion, followed by peripheral blood chimerism analysis at 4-week interval. HSC chimerisms were assessed and secondary transplantation was performed at 16 weeks post induction (Supplementary Fig. 6c). The results confirmed that Med23 depletion impaired both the long-term multilineage repopulation capacity and self-renewal potential of HSCs in a cell intrinsic manner (Fig. 4h–k). In order to exclude the possibility that 4 weeks is not enough for the recovery from transplantation, we started poly(I:C) injection at 12-week post-transplantation, similar results were observed (Supplementary Fig. 7a). Furthermore, competitive limiting dilution experiments were performed to measure the frequency of HSCs. in consistent with our previous results, Med23-deficient mice had lower frequency of HSCs (1 in 496,000) compared with WT mice (1 in 59,000; Supplementary Fig. 7b).

To rule out the possibility that the self-renewal defect seen in *Med23*-deficient HSCs resulting from the defective homing capacity or niche retention of *Med23*-deficient HSCs, we performed HSC homing assay and niche retention assay. The ability of *Med23*-deficient HSCs to be retained in the BM niche was examined by nonablative transplant whole BM cells from WT mice into control or *Med23*-deficient mice[25] (Supplementary Fig. 7c). Interestingly, the niche retention capacity of *Med23*-deficient mice was normal (Supplementary Fig. 7d), indicating that *Med23*-deficient HSCs retained normal niche retention capacity. To assess the ability of hematopoietic stem and progenitor cells (HSPCs) to home to the BM niche, CFSE-labeled WT and *Med23*-deficient HSPCs (LSK) were intravenously injected into the lethally irradiated mice[26]. Notably, no difference was observed in the ability of *Med23*-deficient HSPCs to home to the BM compared to WT controls (Supplementary Fig. 7e), indicating that HSCs retained normal capacity of homing and niche retention in the absence of Med23 deletion. Moreover, the differentiation bias towards myeloid lineage was reproducible under competitive transplantations and whole BM transplantations (Fig. 4l, m), confirming that *Med23*-deficient HSCs harbored enhanced myeloid potential. Collectively, these data suggested that Med23 was cell-autonomously required for the self-renewal capacity of HSCs by controlling the myeloid potential of HSCs.

**Med23-deficient HSCs express myeloid-related genes.** To further reveal the molecular mechanism of Med23 in controlling myeloid potential of HSCs, high-throughput sequencing was performed to analyze the transcriptomes of WT and *Med23*-deficient HSCs (CD150⁺CD34⁻CD48⁻ LSK). Gene-set enrichment analyses (GSEA)[27] were then used to interpret the gene expression profiles in WT and *Med23*-deficient HSCs. Using gene set from myeloid-restricted progenitors defined by Sanjuan-Pla et al.[28] revealed that genes associated with myeloid (PREGM) and MkP programming were enriched in *Med23*-deficient HSCs compared to WT controls (Supplementary Fig. 8a). Of the myeloid-signature genes, 64.7% showed higher expression in Med23-deficient HSCs compared with WT HSCs (Supplementary Fig. 8b). Particularly, in comparison to the WT control, *Med23*-deficient HSCs upregulated certain myeloid differentiation-related genes (e.g., Cebpa, S100a4, S100a6, CD68, Ikzf2), whereas Egr1 a well-known gene that played critical roles in suppressing the activation and differentiation of HSCs was downregulated[6] (Supplementary Fig. 8e). These data indicated that *Med23*-deficient HSCs had expression of myeloid-signature genes.

We then checked genes responsible for HSC self-renewal in *Med23*-deficient HSCs compared with WT controls. As expected, the HSC self-renewal signature genes generated through subjecting the genes from HSC by those from MPP1 defined by Cabezas-Wallscheid N et al.[21] were enriched in WT HSCs but not *Med23*-deficient HSCs (Supplementary Fig. 8c). Of the self-renewal signature genes, 67.1% showed lower expression in Med23-deficient HSCs compared with WT HSCs (Supplementary Fig. 8d). Specifically, genes (e.g., Hif, Ly6a, H19) that played critical roles in HSC maintenance were downregulated[29,30] (Supplementary Fig. 8e).

To further investigate how Med23 regulates transcription, assay for transposase accessible chromatin with high-throughput sequencing (ATAC-seq)[31] was performed to assess the chromatin accessibility upon Med23 deletion. Although loss of Med23 did not affect genome-wide chromatin status (Supplementary Fig. 8f), the locus of *Cebpa*, a key myeloid differentiation gene that was upregulated in Med23-deficient HSCs, was open and more accessible in Med23-deficient HSCs compared with WT HSCs (Supplementary Fig. 8g), indicating that Med23 deletion drove HSCs to express myeloid-related genes prematurely.

As RNA-seq based on population reflects averaged level of different expression genes, single-cell RNA sequencing (scRNA-seq)[32] was performed on both WT and Med23-deficient HSCs to investigate their transcriptomes at single-cell level (Fig. 5a and Supplementary Fig. 9a). Firstly, we introduced GSEA method to compare the lineage programs between WT and Med23-deficient HSCs, using previously identified gene sets[33,34] specific for megakaryocyte/erythroid progenitors (PreMegE genes), granulocyte/macrophage progenitors (PreGM genes), common lymphoid progenitors (CLP genes), and lymphoid-primed progenitors. These results clearly showed a relatively more enrichment of PreMegE- and PreGM- specific genes in Med23-deficient HSCs, with much lesser enrichment of CLP- and lymphoid-specific genes (Fig. 5b). Consistent with our bulk RNA-seq data, the scRNA-seq data confirmed that loss of Med23 initiated a myeloid-transcriptional program in HSCs.

Next, we analyzed the differentially expressed genes between WT and Med23-deficient HSC. Although loss of Med23 resulted in impaired self-renewal in KO mice, Med23-deficient HSCs were in general similar to WT HSCs on the level of the whole

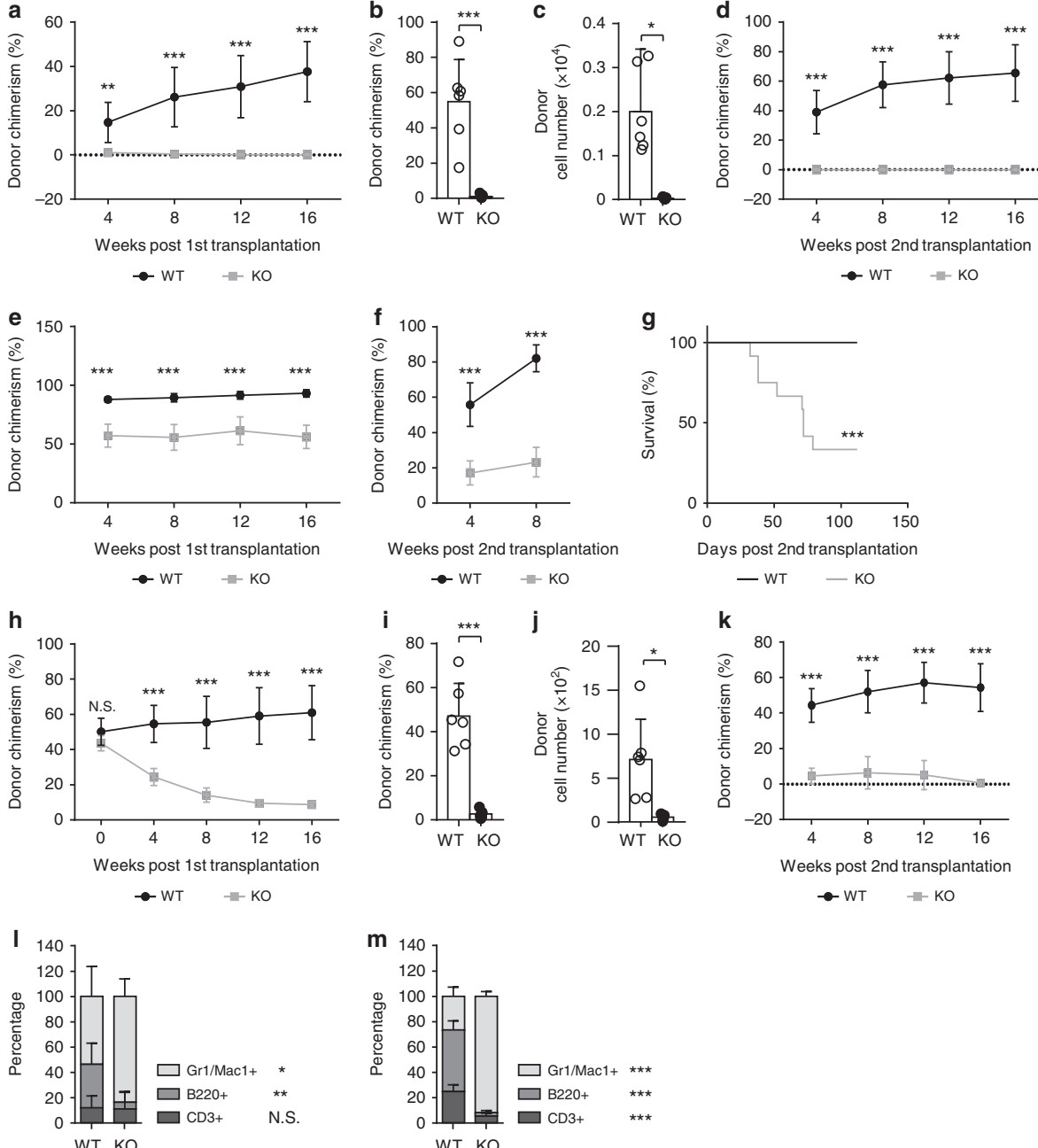

**Fig. 4** Impaired self-renewal potential in med23-deficient HSCs. **a** Donor chimerism of PBMC in primary recipients transplanted with 50 HSCs (CD45.2) from WT or KO mice, along with 500,000 bone marrow cells (CD45.1) (WT, $n = 6$; KO, $n = 7$). **b, c** Chimerism (**b**) and absolute cell number (**c**) of donor HSCs in primary recipients at 16 weeks post-transplantation (WT, $n = 6$; KO, $n = 5$). **d** Donor chimerism of HSCs in secondary recipients transplanted with 2,000,000 bone morrow cells from primary recipients, described in **a** (WT, $n = 11$; KO, $n = 9$). **e** Donor chimerism of PBMC in primary recipients transplanted with 2,000,000 total bone marrow cells (CD45.2) from WT or KO mice (WT, $n = 9$; KO, $n = 8$). **f** Donor chimerism of PBMC in secondary recipients transplanted with 2,000,000 bone morrow cells from primary recipients, described in **e**. **g** Kaplan–Meier survival curve of secondary recipients, described in **f** ($n = 12$). **h** Donor chimerism of PBMC in primary recipients transplanted with 1,000,000 bone marrow cells (CD45.2) of WT and KO mice, along with 1,000,000 bone marrow cells (CD45.1) followed by poly(I:C) administration at 4 weeks post-transplantation (WT, $n = 8$; KO, $n = 7$). **i, j** Chimerism (**i**) and absolute cell number (**j**) of donor HSCs in primary recipients at 16 weeks post-transplantation (WT, $n = 6$; KO, $n = 5$). **k** Donor chimerism of PBMC in secondary recipients transplanted with 2,000,000 bone morrow cells from primary recipients, described in **h** (WT, $n = 8$; KO, $n = 9$). **l** Percent of myeloid, B and T cells in peripheral blood of recipient mice, described in **a** at 16 weeks post-transplantation ($n = 5$). **m** Percent of myeloid, B and T cells in peripheral blood of recipient mice, described in **e** at 16 weeks post-transplantation (WT, $n = 8$; KO, $n = 9$). The data are means ± S.D., for all panels: *$p < 0.05$, **$p < 0.01$, ***$p < 0.001$ by Student's $t$-test, N.S. no significance

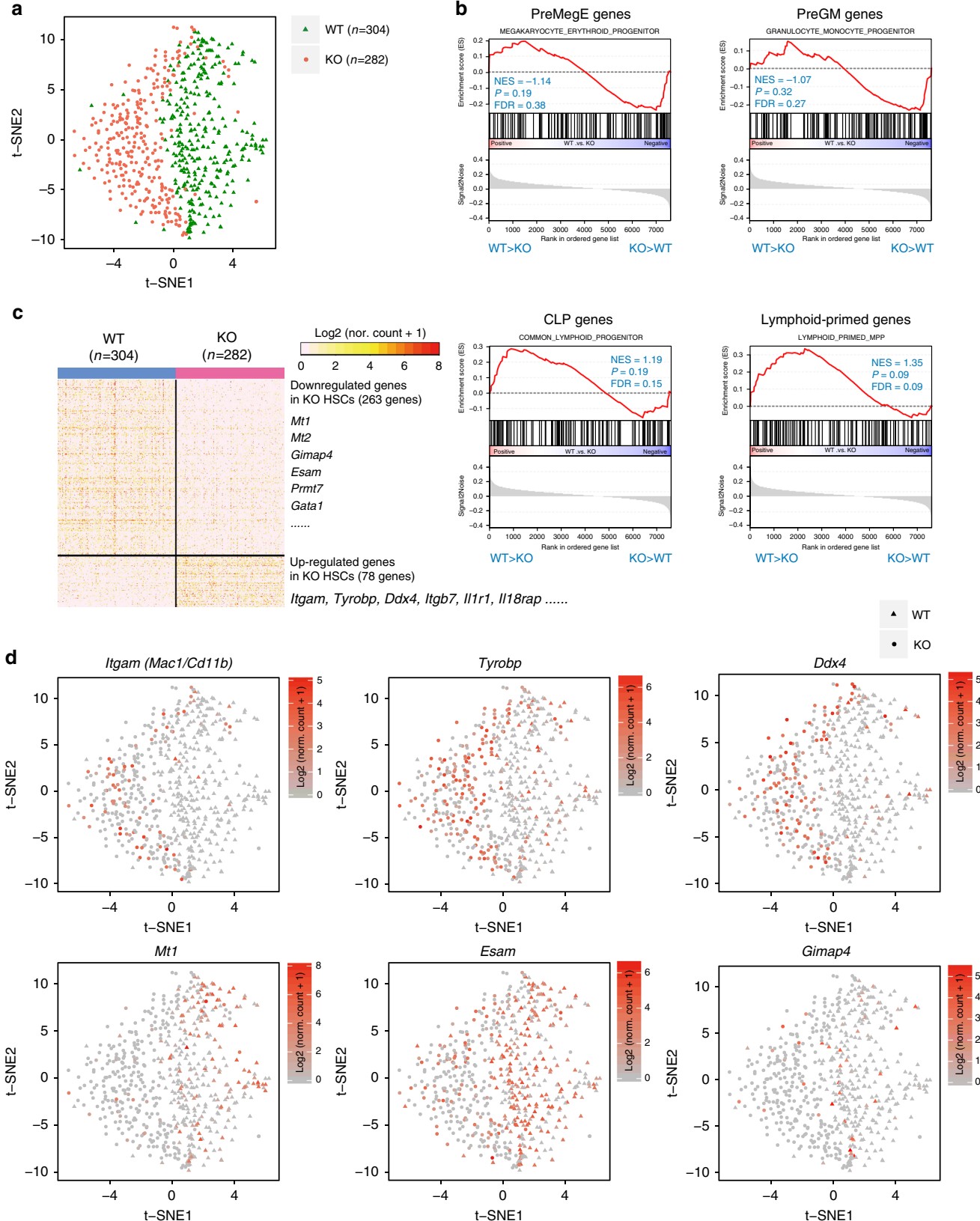

transcriptomes (Supplementary Fig. 9b), and they also expressed HSCs-specific genes such as c-Kit, Sca1, and Cd34, and maintained in G0 or G1 cell-cycle stage marked by Ki67 (Supplementary Fig. 9c). However, there were 78 upregulated genes and 263 downregulated genes in the Med23-deficient HSCs

(Fig. 5c and Supplementary Data 1), Specially, genes that were reported to be involved in myeloid differentiation[35], such as Itgam ($P = 3.79 \times 10^{-14}$, $\chi^2$-test), Tyrobp ($P = 4.03 \times 10^{-13}$), and Ddx4 ($P = 8.99 \times 10^{-16}$), were significantly more expressed in Med23-deficient HSCs (Fig. 5d). And genes were involved in

**Fig. 5** Single-cell RNA sequencing reveals the myeloid-primed signature in Med23-deficient HSCs. **a** t-Distributed Stochastic Neighbor Embedding (t-SNE) plots of WT and KO HSCs. Each green triangle indicates a single WT HSC (total no. = 304), while each coral circle indicates a single KO HSC (total no. = 282). **b** Gene-set enrichment analysis (GSEA) of myeloid or lymphoid signature genes comparing WT with KO HSCs ($n = 304$ for WT HSCs, $n = 282$ for KO HSCs). "NES", "P", and "FDR" stand for Normalized Enrichment Score, Nominal p-value and FDR q-value, respectively. **c** Heatmap of differentially expressed genes between WT and KO HSCs. **d** Expression of representative genes exhibited on the t-SNE plots. Itgam (also known as Mac1/Cd11b), Tyrobp, and Ddx4 were upregulated in KO HSCs, while Mt1, Esam, and Gimap4 were downregulated in KO HSCs. Normalized counts of each gene in single cells were used

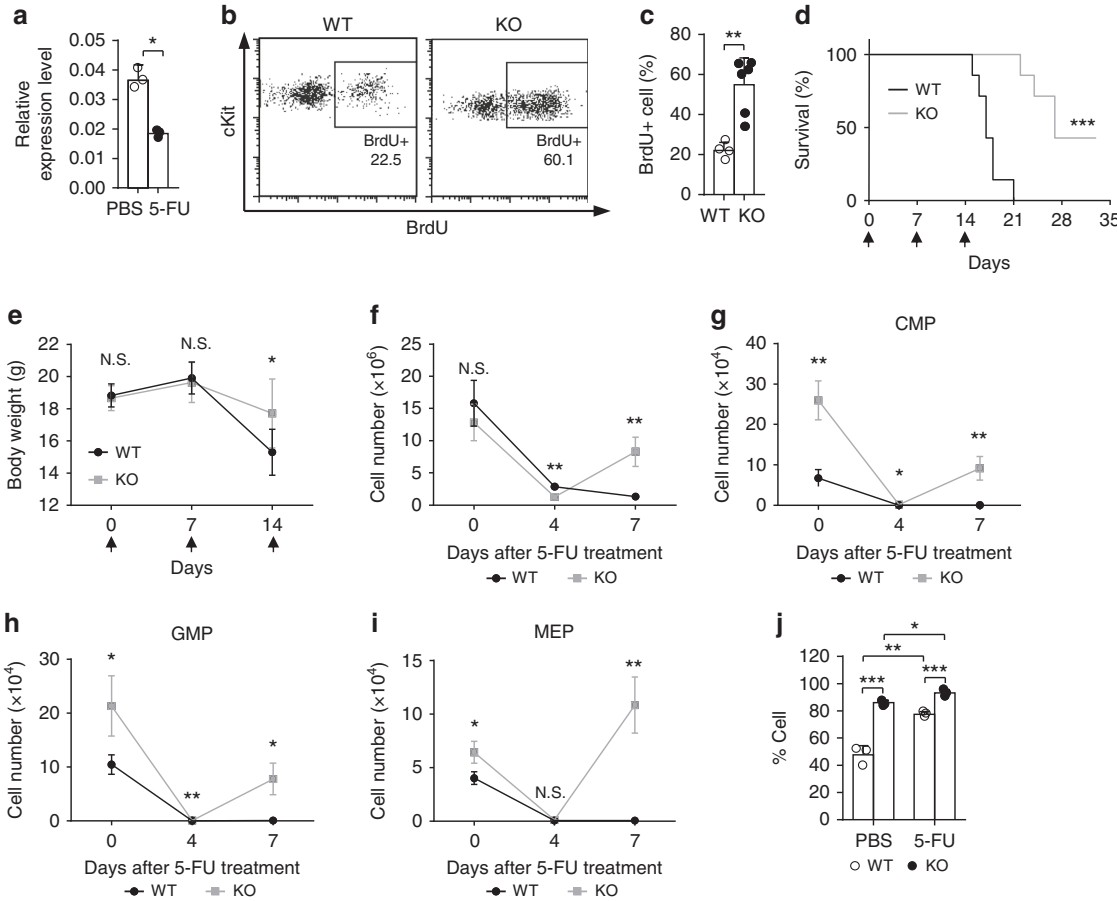

**Fig. 6** Shorter recovery period and better survival of Med23 KO mice under myeloablative stress. **a** Med23 expression in HSC (CD150$^+$CD34$^-$CD48$^-$Lin$^-$Sca1$^+$) isolated from WT mice at 5 days post PBS or 5-FU injection ($n = 3$). **b, c** Representative dot plots (**b**) and percentages (**c**) of BrdU incorporated HSCs in WT and KO mice (WT, $n = 3$; KO, $n = 4$). **d** Kaplan–Meier survival curve of WT and KO mice at different time points after serial 5-FU injection. Arrow shows the time points for 5-FU injection ($n = 7$). **e** Body weights of WT and KO mice at different time points after serial 5-FU injection. Arrow shows the time points for 5-FU injection ($n = 7$). **f** Total bone marrow cells in WT and KO mice after single 5-FU injection ($n = 3$). **g-i** Absolute cell number of CMPs (**g**), GMPs (**h**), and MEPs (**i**) in WT and KO mice at different time points after single 5-FU injection ($n = 3$). **j** Percent of CD41$^+$ cells in HSCs (CD34$^-$CD150$^+$CD48$^-$Lin$^-$Sca1$^+$) from WT and KO mice at 7 days after single 5-FU or PBS injection ($n = 3$). The data are means ± S.D., for all panels: *$p < 0.05$, **$p < 0.01$, ***$p < 0.001$ by Student's t-test, N.S. no significance

lymphopoietic activity[36–38], such as Mt1 ($P = 2.2 \times 10^{-16}$, $\chi^2$-test), Esam ($P = 2.2 \times 10^{-16}$), and Gimap4 ($P = 4.13 \times 10^{-5}$), were significantly less expressed in Med23-deficient HSCs (Fig. 5d). Collectively, these results suggested that Med23 served as a gatekeeper of the myeloid potential of HSCs by suppressing the expression of the myeloid-specific genes and maintaining the normal HSCs program.

**Med23 deletion confers HSCs a better recovery under stress.** Though Med23-deficient HSCs showed impaired HSC self-renewal potential under transplantation, the enhanced myeloid potential might confer some benefits to HSCs when countering physiological stresses. To address this hypothesis, the expression

pattern of Med23 in HSCs was assessed under normal and 5-FU induced myeloablative stress. Interestingly, Med23 was significantly downregulated after 5-FU treatment compared with WT controls (Fig. 6a), highlighting the possibility that Med23 downregulation was required for myeloid recovery under stress conditions. Moreover, when we used the cytotoxic Bromodeoxyuridine (BrdU) to induce HSC activation[39], Med23-deficient HSCs incorporated larger amounts of BrdU compared with control HSCs after 5-day chase (Fig. 6b, c), though Med23-deficient HSCs showed comparable incorporation of BrdU to WT control during short term exposure (supplementary Fig. 4e). These data suggested that Med23 deletion facilitated HSCs activation. We then tested whether Med23 deletion would lead to

better survival of the myeloablated mice under serial 5-FU treatment. As expected, *Med23*-deficient mice showed better survival under serial 5-FU treatment (Fig. 6d). Further analysis revealed that the body weight of *Med23*-deficient mice reduced to a much less level compared with WT controls at 14 days post serial 5-FU treatment (Fig. 6e), which may due to the faster recovery of whole BM cells in *Med23*-deficient mice (Fig. 6f and Supplementary Fig. 10a).

To further elucidate the mechanism that Med23 deletion improved the recovery and survival of the myeloablative mice, myeloid lineage cells were quantitated at different time points after single 5-FU injection. Consistent with the myeloablative function of 5-FU, both WT and *Med23*-deficient myeloid lineage cells were reduced at day 4 post 5-FU treatment (Supplementary Fig. 10b). Notably, *Med23*-deficient HSCs showed an enhanced recovery of the myeloid lineage cells at day 7 post 5-FU treatment (Supplementary Fig. 10b). These findings inspired us to investigate the hematopoietic progenitors in *Med23*-deficient mice. Interestingly, all the myeloid-bias progenitors (CMPs, GMPs, MEPs) in *Med23*-deficient mice were significantly increased at day 7 post 5-FU treatment compared to WT controls (Fig. 6g–i), which was consistent with the tendency seen in the lineage cells. These findings suggested that the *Med23*-deficient HSCs lowered the threshold of activation and harbored enhanced myeloid differentiation potential, thus accelerating the recovery of the myeloid lineage under myeloablative stress.

Finally, we then checked the CD41$^+$ HSCs proportion within *Med23*-deficient HSCs. Interestingly, the proportion of CD41$^+$ HSCs within WT controls were dramatically increased after 5-FU treatment (Fig. 6j), suggesting that WT HSCs may upregulate the expression of CD41, which *Med23*-deficient HSCs was done even under steady state. Altogether, we concluded that Med23 served as a gatekeeper of the myeloid potential of HSCs and Med23 deletion conferred HSCs a better recovery under myeloablative stress.

## Discussion

The mechanism by which HSCs initiate a rapid activation under physiological stresses is a long-standing question in the field, and the key factors that control the activity of HSCs during activation remain largely unknown. Here, we show that Med23 is a bona fide transcriptional regulator that controls the myeloid potential of activated HSCs. *Med23*-deficient HSCs undergo myeloid-biased differentiation with impaired self-renewal, resulting in lymphocytopenia. Moreover, Med23 plays essential roles in maintaining the stemness genes and suppressing the myeloid lineage genes, and hence prevents HSCs from being the myeloid potential and loss of self-renewal capacity. Physiologically, Med23 is downregulated in HSCs under myeloablative stress and *Med23*-deficient HSCs leads to enhanced myeloid recovery and better survival after serial 5-FU treatment. Altogether, our findings identified Med23 as a gatekeeper of the myeloid potential of HSCs.

Our previous findings have suggested that Med23 controls the activation threshold of T-cell by maintaining the expression of several negative regulator of T-cell activation[15]. In this study, we found that Med23 suppresses the expression of myeloid lineage genes (e.g., Cebpa, S100a4, S100a6, CD68, Ikzf2) and maintains the expression of the differentiation inhibition gene Egr1[6]. Upon myeloablative stresses, HSCs physiologically downregulate Med23, which enables HSCs to enter primed state (CD41$^+$ HSC), thus facilitating HSC activation and myeloid differentiation. After recovery, Med23 expression is probably restored, and lead to the re-expression of stemness genes, which will help HSCs return to the quiescence state. Altogether, Med23 may play a unique role in the threshold setting to prevent cell state change by repressing the expression of activation- and differentiation-related genes.

The mediator complex is composed of more than 20 subunits. Some subunits are essential for maintaining the stability of the entire mediator complex, while others bridge the mediator complex to the different machineries to facilitate the precise transcriptional function of the mediator complex[13,14]. In this study, we found that loss of Med23 does not cause destabilization of the mediator complex. Interestingly, deletion of Med12, another subunit of the mediator complex, also preserve the intact mediator complex, suggesting Med23 and Med12 may facilitate the interaction of mediator complex to other cofactors[40]. Indeed, Med12 bridges the mediator-bound promoters of HSC-specific genes to relative enhancers to maintain the hematopoietic transcriptional programs. According to our study, Med23 serves distinct functions in HSCs, which maintains the stemness genes and suppresses myeloid differentiation genes. Moreover, unlike Med12, loss of Med23 has no effect on animal survival. The studies of Med12 and Med23 in hematopoiesis suggest that different subunits of the mediator complex play distinct roles in regulating the transcription progresses, thus leading to different phenotypes.

It has been reported that primed-state HSCs are much easier to be activated in response to physiological stresses[4,41,42] and have enhanced myeloid potential and impaired self-renewal[43]. In this study, Med23-deficient HSCs are more rapidly activated in response to myeloablative stresses, such as 5-FU treatment and contribute to a rapid recovery from the myeloablative stress. Therefore, HSCs may enter a ready state in the absence of Med23. Besides, considering that Med23-deficient HSCs have higher expression of CD41 and have enhanced myeloid-biased differentiation, we reason that in HSCs, Med23 deletion opens a gateway for myeloid fate decisions, generating myeloid-primed HSCs. Notably, Med23 deficiency results in loss of functional HSCs and Med23-deficient HSCs are phenotypic HSCs, which suggests that Med23 is also probably required for self-renewal of HSCs. In the myeloablative stress, HSCs can downregulate the expression of Med23, which primes HSCs for myeloid potential in order to replenish the myeloid cells, and this process is revisable resulting in maintaining self-renewal of HSCs. However, although Med23-deficient HSCs may enter to the primed state of HSCs with the rapid recovery from the myeloablative stress, Med23-deficient HSCs can not restore expression of Med23 resulting in loss of functional HSCs. Altogether, we reason that Med23 serves a gatekeeper of myeloid potential and prevent loss of self-renewal in HSCs.

Normal HSCs can balance their self-renewal and differentiation to sustain life-long hematopoiesis, whereas imbalance between HSC self-renewal and differentiation will result in HSC exhaustion owing to the enhanced differentiation or pancytopenia due to reduced differentiation. In this study, though Med23-deficient HSCs are highly correlated with WT HSCs, they upregulate myeloid-specific genes but repress stemness and lymphoid lineage genes at single-cell level, leading to enhanced myeloid differentiation potential at the expense of self-renewal capacities and lymphoid differentiation potential. The data highlight that Med23 is one of the key transcriptional regulators that balance the self-renewal capacity versus myeloid differentiation potential though controlling the myeloid-primed state of HSCs. When encountering acute stresses (e.g., 5-FU treatment), *Med23*-deficient HSCs can confer the host a speedy recovery and better survival under acute stresses. In light of this, therapies can be developed by targeting Med23 and its downstream targets to set the HSC into the myeloid potential, which helps to improve the recovery of the hematopoietic system under acute stresses.

Acute myeloid leukemia (AML) is a clinically devastating disease. One hallmark of AML is that the leukemic blast gains the ability to self-renew and is arrested at an early stage of differentiation[44]. This phenomenon raises the hypothesis that targeting the factors that halt the leukemic blast from differentiation may become promising therapies. Therefore, investigating the molecular wiring of HSC differentiation is important, because it may be possible to cure leukemia by targeting this critical process. Previous research have identified Med23 as an oncogene and the transcriptional shifts driven by Med23 mutation contributes to Ras addiction during lung carcinogenesis[19] and Med23 mutation results in intellectual disability because of dysregulation of IEG expression[18]. Here, in this study, we show that Med23 deletion leads to impaired HSC self-renewal accompanied with enhanced myeloid differentiation, highlighting the possibility that Med23 may serve similar function in leukemic stem cells (LSCs) and leukemia blast. If so, targeting Med23 in LSCs or leukemia blasts may trigger the differentiation program and leads to the exhaustion of LSCs and leukemia blasts. Further investigations should be conducted to elucidate the underlying mechanisms and to determine whether Med23 can serve as a potential target for defining new approaches to leukemia therapy.

## Methods

**Mice**. The Med23 floxed mice have been described in our recent paper[15]. The Mx1-cre mice (strain: B6.Cg-Tg(Mx1-cre)1Cgn/J) transgenic mice, UBC-cre/ERT2 mice and CD45.1 mice (strain: B6.SJL-PtprcaPepcb/BoyJ) have been described[20,45]. All mice were bred and housed in specific pathogen-free conditions. All mice were genotyped by PCR and genotyping primer sequences were available in Supplementary Table 1. Mx1-cre–induced gene deletion was done by intraperitoneal injection of poly(I:C) (300 μg per mouse) four times at 2-d intervals. ERT2–induced gene deletion was done by intraperitoneal injection of tamoxifen (1 mg per mouse) five times for 5 consecutive days. All mice were genotyped using PCR analysis before experimentation. 5-FU was injected intraperitoneally into mice (150 mg per kg body weight) every week for three times at 3 weeks after poly(I:C) injection. For animal studies, all animals were randomly allocated to experimental groups and processed. And we were not blinded to the group allocation during experiments and outcome assessment. All animal experiments were approved by the Institutional Animal Care and Use Committee of the Shanghai Institutes for Biological Sciences, Chinese Academy of Sciences.

**Transplantation assays**. For BM transplantation, all recipient mice had been lethally irradiated using X-ray with two doses of 5.40 Gy (total 10.8 Gy), delivered at least 2 h apart except as otherwise described. For total BM transplantation, 2,000,000 whole BM cells were transplanted into recipient mice (CD45.1+) by tail vein injection. For second transplantation, 2,000,000 whole BMs from 1st receipt mice were transplanted into recipient mice (CD45.1+). Survival was checked every week. For competitive BM transplantation, 50 sorted HSCs (CD34−CD150+CD48 − LSKs) (CD45.2+) were transplanted into recipient mice along with 500,000 competitive BM cells derived from non-irradiated recipient mice (CD45.1+). Donor chimerisms of peripheral blood were assessed every 4 weeks. BMs were analyzed at 16 weeks after transplantation. And 2,000,000 total BM cells were transplanted into recipient mice for 2nd transplantation. Peripheral blood was analyzed by FACS every 4 week for 4 times. For another competitive transplantation, total BM cells from CD45.1+ competitor and CD45.2+ mice were mixed at a ratio of 1:1 and a total of 2,000,000 cells were injected intravenously into lethally radiated recipient mice. At 4-week post-transplantation, donor chimerism was assessed and Med23 deletion was achieved by poly(I:C) administration. After deletion, the analysis was performed as same as above.

For competitive limiting dilution assay, three doses of donor BM cells ($5 \times 10^4$, $1 \times 10^5$ and $2 \times 10^5$) along with $5 \times 10^5$ recipient BM cells (CD45.1) were transplanted into irradiated mice. After 16 weeks, donor chimerisms of recipient mice were analyzed by FACS.

For HSCs BM retention assay, 40,000,000 BM cells from non-irradiated CD45.1+ were transplanted into WT or KO CD45.2+ mouse. BM cells were analyzed at 12-week post-transplantation. For homing assay, 10,000 sorted HSPCs (LKSs) were labeled with CFSE and transplanted into lethally irradiated CD45.1+ mice. After 16 h, the BM cells of recipient mice were analyzed by FACS.

**Flow cytometry and cell sorting**. BM cells were obtained by flushing tibias and femurs from experimental mice, followed by red blood cell lysis before filtration. Surface staining was carried out in HBSS with 0.1% BSA staining buffer for 40 min or followed by streptavidin for 20 min[46]. The information of antibodies and reagents was available in Supplementary Data 2 The following antibodies were used

to define lineage+ cells: anti-CD3e (145-2C11), anti-CD4 (GK1.5), anti-CD8 (53-6.7), anti-Gr1 (RB6-8C5), anti-TER119 (TER119), and anti-B220 (RA3-6B2). The following additional antibodies were used to define LT-HSCs, ST-HSCs, MPPs, and HPCs (LKs): anti-cKit (2B8), anti-Sca1 (D7), anti-CD48 (HM48-1), anti-CD150 (TC15-12F12.2), anti-CD135 (A2F10.1), and anti-CD34 (RAM34). For identifying CMPs, GMPs, and MEPs, anti-CD16/32 (93) were used. To measure donor-derived chimerism, peripheral blood from recipients was obtained by the tail vein-bleeding method and prepared as previously described[20], the following antibodies were used to assess multilineage reconstitution: anti-CD3e (145-2C11), anti-B220 (RA3-6B2), anti-Gr1 (RB6-8C5), anti-Mac1 (M1/70), anti-CD45.1 (A20), and anti-CD45.2 (104). All antibodies were from BD Pharmingen, Biolegend, or eBioscience. Cell fluorescence was acquired on a four-laser BD LSRFortessa II and was analyzed with FlowJo software. For cell sorting, lineage depletion was first done according to manufacturer's instruction after cell isolation from BMs. Cell sorting was carried by a BD FACSAria II after surface staining. Sorted cell purity was over 90%.

**Immunoprecipitation and western blot**. Cells were lysed on ice for 30 min in lysis buffer with protease inhibitor mixture, and the lysates were cleared by centrifugation. The resulting supernatants were immunoprecipitated with Pol II Abs at 4 °C overnight, then added Protein A/G PLUS-Agarose (sc-2003; Santa Cruz Biotechnology) at 4 °C for 4 h. After washing, 5× sample loading buffer was added to the immunoprecipitates. Samples were subjected to immunoblot analysis with the indicated Abs and HRP-conjugated secondary Ab (Santa Cruz Biotechnology). The following antibodies were used: Santa Cruz: Pol II(A-10); BD Pharmingen: Med23(550429); Novus Biologicals: MED24 (NB100-74599); Lianke Biotechnology: beta-actin (70-ab008-100).

**BrdU incorporation and apoptosis analysis**. For proliferation assay, BrdU (100 mg/kg body weight; Sigma) was injected intraperitoneally 18 h before been sacrificed. For long-term stress, BrdU was injected intraperitoneally and mice were administrated with BrdU drink water (0.8 mg/ml) for 5 days changed every 2 days before killing. BrdU staining was performed using the APC BrdU Flow Kit (BD Pharmingen). Annexin V (Biolegend) and DAPI (Cell Signaling Technology) were used in apoptosis assays.

**In vitro differentiation with OP9 cell line**. OP9 cell line has been described[47]. FACS-sorted HSCs (CD150+CD48- LSKs) were cultured on mouse OP9 stromal cells and supplemented with hematopoietic cytokines (50 ng/ml SCF, 10 ng/ml FLT3L, 10 ng/ml IL-7). After being cultured for 12 days, the cells were harvested by mechanical pipetting (hematopoietic cells) for flow cytometry analysis.

**RNA-seq**. HSCs (CD34−CD150+CD48- LSKs) (~2000cells) were isolated by cell sorting (double sort, purity > 95%). RNA was extracted, purified, and checked for integrity using an Agilent Bioanalyzer 2100 (Agilent Technologies). Libraries were generated for sequencing using the TruSeq RNA sample preparation kit (Illumina). Libraries were sequenced using an Illumina HiSeq. 2500 sequencer. All of the above processes were performed at Shanghai Biotechnology Corporation, Shanghai, China.

**ATAC-seq and analysis**. ATAC-seq was performed according to the protocol[31]. Briefly, 5000 HSCs (CD150+CD48− LSK) were sorted and treated by Tn5 (Illumina Kit) for 30 min at 37 °C and DNA was isolated for building library. Raw data from the sequencer were first underwent quality control using FastQC. Next, reads are treated using trim_galore. Illumina adapter sequence and low quality (phred score < 20) at the 3′ end were trimmed. The paired reads were removed if any of the two reads does not meet the minimum length (20 bp).

Reads were aligned to mm10 using BOWTIE2 using the parameter –X 2000, which ensures fragments up to 2 kb were allowed to align. For all data files duplicates and mitochondrial reads were removed using Picard.

We used MACS2 to call ATAC-seq peaks, using the–nomodel option. To quantitively compare the difference of peaks between KO and WT, we make a consensus peaks through merging the overlapped peaks of KO and WT. For each sample, we count the reads hitted on the consensus regions of each peak using featureCount. Based on the counts, we calculate the ratio of the difference between KO and WT.

**Quantitative real-time PCR analysis**. Total RNA was extracted from FACS-sorted HSCs using the Quick-RNA MicroPrep Kit (Zymo Research), then reverse-transcribed with the HiScript II Q RT SuperMix for qPCR ( + gDNA wiper) Kit (Vazyme). Real-Time PCR was performed using SYBE Green Realtime PCR master Mix (TOYOBO) on a Rotor-Gene Q machine. GAPDH was used as internal control. The primer pairs for the genes examined were listed in Supplementary Table 1.

**Single-cell RNA-seq library construction**. Single mouse HSCs which were isolated by cell sorting were manually picked, lysed, and subjected to first-strand cDNA synthesis by using barcoded oligo-dT primers (Supplementary Data 3)[48–50]. Then, second-strand cDNA were synthesized, amplified, and fragmented by

sonication (Covaris). Final RNA-seq library was prepared by using the KAPA Hyper Prep Kit (KAPA Biosystems). Libraries were checked, pooled, and sequenced on Illumina HiSeq 4000 platform (paired-end 150-bp reads).

**Single-cell RNA-seq data analysis**. Raw reads were separated by cell barcodes, trimmed, and aligned to the mm9 mouse transcriptome and de-duplicated by UMIs information[51]. Then, the transcript copy numbers of each gene were counted and used for the subsequent analysis. Single cells with less than 30% mapping efficiency, 2000 detected genes and 20,000 transcripts were excluded. Genes with averaged counts above 0.2 among all the retained 586 single cells were used, and the transcripts were normalized. The t-Distributed Stochastic Neighbor Embedding (t-SNE) analysis were performed by using the "Rtsne" package in the R software. GSEA were conducted following the recommended instructions, and PreMegE genes (System no. M7007), PreGM genes (System no. M7055), CLP genes (System no. M8877), and Lymphoid-primed genes (System no. M8829) were used for the analysis (http://software.broadinstitute.org/gsea/index.jsp).

**Statistical analysis**. All data are expressed as means ± S.D. and two-tailed unpaired Student's $t$-test were used to determine statistical significance. For all experiments: $*p < 0.05$, $**p < 0.01$, $***p < 0.001$.

## Data availability

RNA-seq and ATAC-seq data have been deposited in NCBI's Gene Expression Omnibus and are available through GEO Series accession number GSE112359 and single-cell RNA-seq data are available from GSE112008. The authors declare that all the data supporting the findings of this study are available within the article and its supplementary information files and from the corresponding authors upon reasonable request.

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

## Acknowledgements

We thank Baojin Wu and Guoyuan Chen for the animal husbandry and Wei Bian for the support of cell sorting. This work was supported by National Natural Science Foundation of China (31530021, 31621003, 31500717, and 31501193), the Strategic Priority Research Program of the Chinese Academy of Sciences (Grant XDB19000000), National Basic Research Program of China (2013CB835300), the Youth Innovation Promotion Association of Chinese Academy of Sciences, and the China Postdoctoral Science Foundation (2015M581672).

## Author contributions

X.C. and J.Z. contributed equally to this work. X.C. and J.Z. designed, performed, and analyzed the experiments. H.L., G.S., H.X., H.S., Y.D., Y.C., X.Z., and Y.W. performed experiments. Y.L. and P.H. analyzed the ATAC-seq. C.G. and F.G. performed single-cell RNA seq and analyzed data. X.C., J.Z., F.G., and X.L. prepared the manuscript. X.L. conceptualized the research and directed the study.

## Additional information

**Competing interests:** The authors declare no competing interests.

