## [Peer Review File · Nature Communications]

Editorial note:

Parts of this peer review file have been redacted as indicated to remove third-party material where no permission to publish could be obtained.

Reviewers' comments:

Reviewer #1 (Remarks to the Author):

The Authors have presented a nice study characterizing the role of Med23 in HSC differentiation. Using Mx-Cre mediated conditional deletion of Med23 they show that Med23-deficiency leads to lymphocytopenia while myeloid lineage cell differentiation remains unaffected by loss of Med23. In addition, loss of Med23 causes a cell-autonomous self-renewal defect in HSCs, evidenced by impaired hematopoietic reconstitution capacity.

This is (together with a recent MED12 publication) one of the first papers to dissect the role of the Mediator complex in adult hematopoiesis. To strengthen the manuscript the authors should make an effort to expand further on their current findings.

Main comments:

What happens to the frequency of various subsets of myeloid cells and are these all differentiating normally (Granulocytes, Neutrophils, monocytes and macrophages...).

What is the expression pattern of Med23 during hematopoiesis? Is Med23 more highly expressed in the lymphoid compartment compared to myeloid lineage cells? And is there differential expression of Med23 in the hematopoietic stem and progenitor cell compartment?

If Med23 is overexpressed, does this promote a shift from myeloid to lymphoid differentiation bias in HSCs/HSPCs? Could the authors address this using a retrovirus to introduce this into wild-type HSPCs for in vitro or in vivo differentiation assays.

At the protein level, does loss of Med23 cause destabilization of the mediator complex only in the lymphoid compartment?

From a mechanistic perspective it would be important to complement transcriptomic studies with ATAC-seq or ChIP for Med23 (or other members of the mediator complex such as Med1) to understand locus specific and directly regulated genes affected by loss of Med23.

Minor comments:

On page 9: Twice the authors have referred to RNA sequencing results in Uhrf1-deficient FL-HSPCs, which are not studied in this manuscript.

The discussion and many conclusions in the results section are repetitive.

The importance and known role of Med23 in the context of the entire mediator complex should be discussed with its relevance to the current findings of its role in hematopoietic development. The role of Med23 loss of function in disease and its relevance in hematopoietic malignancies also needs to be further discussed.

Reviewer #2 (Remarks to the Author):

The authors present analyses of the hematopoietic impact of Med23 deletion. The mice develop an interesting phenotype which the authors have interpreted as an increase in myeloid-priming of HSCs.

Main comments:

Although the phenotype is striking (expansion of myeloid progenitors, loss of self renewal and loss of lymphoid cells), the current analyses do not allow clear conclusions to be reached. In essence, deletion of Med23 leads to an almost complete loss of functional HSCs as demonstrated in transplantation assays shown in Figure 4. Therefore, both functional and molecular analysis of phenotypic HSCs is irrelevant for HSC biology as these cells lack the HSC-defining property of self-renewal and likely no longer represent HSCs at all but rather some form of progenitor cell.

To demonstrate lineage-bias it is essential to purify single-HSCs or MPPs pre-deletion of Med23 and then subject these cells to in vitro combined myeloid and lymphoid single-cell lineage assays with or without deletion. These experiments are challenging but without such single-cell analyses it is not possible to reach conclusions about lineage biases. Even then, it is crucial to exclude that the loss of lymphoid cells does not relate to a differentiation block in lymphoid development rather than a myeloid bias of multipotent cells. This again requires purification of HSCs pre-deletion of Med23, and then subjecting single cells to a lymphoid promoting culture and deleting at an early time point when full lymphoid commitment has occurred. If the hypothesis is correct then there should be no impact of Med23 loss in this assay.

Specific comments:

1. Med23 deficient mice develop thrombocytopenia, what is the explanation for this? Is Mk-priming of HSCs affected?
2. For quantification of B-cells data for CD19+ should be shown and for myeloid cells Mac1+/Gr1- as well as Mac1+/Gr1+
3. The expansion of CMP and GMP is at odds with the low normal numbers of myeloid cells in the blood. Are these CMPs and GMPs less effective at making mature myeloid cells?
4. Same issue for the increased MEP population despite reduced platelet count and borderline reduced Hgb in these mice.
5. What is the total number of phenotypic LT-HSCs and at what timepoint following deletion are these analyses carried out? The analysis should be a number of weeks (minimum 6 weeks) after poly i:c administration.
6. The data in Fig4E-F appear to show ongoing chimerism when total BM cells and not HSCs are transplanted (unlike 4A and 4H). There is no loss of chimerism with time (unlike 4A and 4H). What is the explanation for this?
7. The use of Mx1-Cre is highly problematic for the observed phenotype. It is quite possible that there is some form of combinatorial effect between poly i:c administration and loss of Med23. An alternative Cre line such as Vav-Cre or a different inducible system should also be used in parallel to exclude a poly i:c effect.
8. Myeloid (and platelet) biased HSCs expand with age. Mice should be aged and analysis carried out to study the impact of Med23 loss on the HSC aging phenotype.
9. What is the survival of primary mice following Med23 deletion? Do mice eventually develop BM failure. Is there any evidence of dysplasia in the myeloid cells?

Reviewer #3 (Remarks to the Author):

Med23, a subunit of the transcriptional mediator complex, plays a role in controlling transcriptional profiles in different cell types. To investigate a potential role for Med23 in hematopoietic stem cells, an inducible mouse model is used to delete Med23 in the hematopoietic compartment. Based on their

data the authors conclude that Med23 serves as a gatekeeper of the myeloid primed state of HSCs. The data presented with the mouse model are interesting, however based on these data, alternative explanations are possible as well. The initial observation in Figure 1 is that mice lacking Med23 in the hematopoietic system show reduced lymphoid cell numbers, but the number of myeloid cells in these mice stays the same. The authors conclude there is a myeloid bias in these mice and continue on this thought. Of course this decrease in lymphoid cells is changing the lymphoid-myeloid ratio in these mice. However, an alternative explanation of this phenotype could be that these mice have a compromised lymphoid compartment in stead of a myeloid bias, as has been shown for many immune-deficient mouse models. Throughout the manuscript the focus is on explaining the myeloid bias, ignoring any possible defect in lymphoid differentiation. In addition, the authors talk about a 'primed state' of HSCs. However the papers they are citing do not necessarily provide evidence for a 'primed state'. Furthermore, the term 'primed state' is used for different HSC populations in different papers, it's not a well-accepted term for a certain subpopulation of HSCs.

Furthermore, there are several open questions/concerns:

- are all the analyses performed at the same timepoint after pIC injection (21 days post last injection)? why was this timepoint chosen? How does the phenotype develop over time? What about the survival of the mice? (Figure 1, 2, 3)
- Have the authors confirmed the Med23 phenotype in a different inducible mouse model which is independent on pIC induced (and thus stress induction) deletion of Med23?
- The panels in Figure 1 should be better labeled. It should be clear from the labeling of the y-axis which cell numbers are shown (myeloid, B cells, NK cells). Were the cell numbers in the peripheral blood also analyzed to get a better overview of overall cell numbers of differentiated cells in the mice?
- Figure 2D&E: did the authors check whether there might be a general increase in the CD34+ signal leading to a 'change' in populations? Why is there no absolute increase in myeloid cells if there is an increase in the number of myeloid progenitors?
- There is quite some controversy in the field whether CD41 marks a subset of HSCs with strong myeloid potential. The level of CD41 in Figure 3A on wt HSCs is very high compared to published data, why?
- Figure 3E would indicate a change in CD48 marker expression upon pIC treatment. Where do the CD50+CD48+ cells go in Med23KO mice? Whether the DN (CD150-CD48-) are the ST-HSCs is not very clear in the field. What happens to the expression of cKit and Sca-1, are there changes, both under homeostasis as well as upon pIC treatment?
- For the stress treatments in Figure 3 both the data under homeostasis as well as upon pIC treatment should be shown.
- Did the authors analyze cell cycle distribution with for example Ki67-Hoechst or shortterm BrdU incorporation (18h) as measurement for proliferation at baseline and upon pIC treatment? The BrdU data shown are longterm incorporation assays which are more of a stress readout.
- Are Med23 KO HSCs cycling more and could this be part of the reason why they perform worse in a transplant assay (Figure 4A)? Did they try to transplant HSCs i.f.? Did the authors perform limiting dilution experiments to analyze the difference in HSC frequency?
- In Figure 4H-K the authors start pIC injection only 4 weeks post transplantation. At this time point there is no good recovery yet in the transplanted mice and reconstitution is not fully established yet. What happens when they would wait until 12 weeks post transplant and then inject with pIC?
- In Figure 4L-M the percentage of cells are shown, not the total nr of cells. The alternative explanation here could again be that the number of B and T cells are reduced due to reduced lymphoid cell production and thus the % of myeloid cells are higher.
- The data analysis of the RNAseq in Figure 5 are very limited and rather one sided. How about a lymphoid signature? Is the myeloid signature the only major difference? In the text they talk about Uhrf1-deficient FL-HSPCs????
- What is expression of Med23 in the different hematopoietic cell populations under homeostasis in

stead of upon 5FU (Figure 6A)?

- Figure S6 is missing several panels: hard to say something about this data without the actual data present.
- In Figure 6 the different populations are analyzed upon 5FU treatment. However 5FU is causing a lot of marker changes, which are still present 4 and 7 days post 5FU. Therefore it's difficult to judge these data. The raw FACS plots would be very useful here.
- The suggestion based on the data shown in Figure 6H is pure speculative, there is no data to back this up.
- In the discussion the authors speculate about what happens to Med23 expression upon recovery from stress and how Med23 deficient HSCs behave during the recovery phase. These are easy hypothesis to test in their model and would increase the message of their story.
- How would Med23 function as a gatekeeper other than the general explanation of it control transcription? A bit more mechanism here would be informative.
- the text would benefit a lot from an English grammar correction. Some of the sentences are difficult to read because of problems with the English.

Reviewer #1 (Remarks to the Author):

The Authors have presented a nice study characterizing the role of Med23 in HSC differentiation. Using Mx-Cre mediated conditional deletion of Med23 they show that Med23-deficiency leads to lymphocytopenia while myeloid lineage cell differentiation remains unaffected by loss of Med23. In addition, loss of Med23 causes a cell-autonomous self-renewal defect in HSCs, evidenced by impaired hematopoietic reconstitution capacity.

This is (together with a recent MED12 publication) one of the first papers to dissect the role of the Mediator complex in adult hematopoiesis. To strengthen the manuscript the authors should make an effort to expand further on their current findings.

We thank the reviewer for his/her positive comments on our study.

Main comments:

What happens to the frequency of various subsets of myeloid cells and are these all differentiating normally (Granulocytes, Neutrophils, monocytes and macrophages...).

We have measured the frequency of neutrophil cells (Gr1⁺Mac1⁺), immature myeloid cells (Gr1^{low}Mac1⁺) and other myeloid cells (Gr1⁻Mac1⁺) in bone marrows according to the reviewer's suggestion. As seen in figure S2C, compared with WT, the frequency of neutrophil cells is decreased and the frequency of immature myeloid cells and the other myeloid cell are increased in Med23-deficient mice, suggesting Med23 might play a role in the developmental processes from myeloid progenitors to mature myeloid cells. We have added an explanation in Page 6.

What is the expression pattern of Med23 during hematopoiesis? Is Med23 more highly expressed in the lymphoid compartment compared to myeloid lineage cells? And is there differential expression of Med23 in the hematopoietic stem and progenitor cell compartment?

We have measured the expression pattern of Med23 in different hematopoietic cells including HSCs and HPCs. As shown in Figure S1A, the relative high expression of Med23 in MEP, B cells and T cells is found among different hematopoietic stem and progenitor cells as well as the lineage cells.

If Med23 is overexpressed, does this promote a shift from myeloid to lymphoid differentiation bias in HSCs/HSPCs? Could the authors address this using a retrovirus to introduce this into wild-type HSPCs for in vitro or in vivo differentiation assays.

We overexpressed Med23 in bone marrow cells from 5-FU treated WT mice. After 2-day culture, transfected cells (GFP⁺), together with Sca1-depletion bone marrow cells, were transplanted into lethally-irradiated mice. After 8 weeks, the chimerism of donor-derived lymphoid cells in peripheral blood was analyzed by flow-cytometry. As shown below Figure 1, no difference is found between control and Med23-overexpression.

Figure 1 Lymphoid differentiation of HSC overexpressed Med23 and control (VEC). Med23 is overexpressed in bone marrow cells from WT mice treated with 5-FU and after 2-day culture, transfected GFP⁺ cells with Sca1-depletion bone marrow cells were transplanted into lethally-irradiated mice. After 8 weeks, lymphoid cells in GFP⁺ peripheral blood cells were analyzed by FACS. (Med23, n=12, VEC, n=13. N.S.: no significance).

At the protein level, does loss of Med23 cause destabilization of the mediator complex only in the lymphoid compartment?

Our previous study reported that CD4-cre induced deletion of Med23 at DP stage in the thymus didn't affect T cell development from DP stage to SP stage¹, suggesting that the mediator complex is stable in T lymphocytes in the absence of Med23. In addition, the paper published in *Cell Research* also proved that the loss of Med23 does not cause destabilization of the mediator complex².

We have investigated the stability of the mediator complex through immunoprecipitation with Pol II antibody in hematopoietic progenitor cells (Lineage⁻cKit⁺ population). Western blotting showed that as a component of mediator complex, the protein level of Med24 was comparable to control in the absence of Med23, indicating that the complex is intact after Med23 depletion (Figure S1E).

From a mechanistic perspective it would be important to complement transcriptomic studies with ATAC-seq or ChIP for Med23 (or other members of the mediator complex such as Med1) to understand locus specific and directly regulated genes affected by loss of Med23.

We thank the reviewer's suggestion. ATAC-seq analysis was performed (Figure S7F,G). Med23 loss didn't affect whole chromatin accessibility (Figure S7F), however, the loci of *Cebpa*, the key myeloid signature gene, are more open in Med23-deficient HSCs than their counterparts (Figure S7G).

Minor comments:

On page 9: Twice the authors have referred to RNA sequencing results in Uhrf1-deficient FL-HSPCs, which are not studied in this manuscript.

Sorry for our mistake. We have corrected this in the revised manuscript (Page 10).

The discussion and many conclusions in the results section are repetitive.

The importance and known role of Med23 in the context of the entire mediator complex should be discussed with its relevance to the current findings of its role in hematopoietic development. The role of Med23 loss of function in disease and its relevance in hematopoietic malignancies also needs to be further discussed.

We agree with the reviewer's points and we have discussed more in the revised version (Page 14-16).

Reference

1. Sun Y, Zhu XY, Chen XF, Liu HF, Xu Y, Chu YJ, et al. The mediator subunit Med23 contributes to controlling T-cell activation and prevents autoimmunity. *Nat Commun* 2014, 5: 5225.
2. Chu YJ, Rosso LG, Huang P, Wang ZC, Xu YC, Yao X, et al. Liver Med23 ablation improves glucose and lipid metabolism through modulating FOXO1 activity. *Cell Res* 2014, **24** (10): 1250-1265.

Reviewer #2 :

The authors present analyses of the hematopoietic impact of Med23 deletion. The mice develop an interesting phenotype which the authors have interpreted as an increase in myeloid-priming of HSCs.

We thank the reviewer for his/her interest in our study.

Main comments:

Although the phenotype is striking (expansion of myeloid progenitors, loss of self renewal and loss of lymphoid cells), the current analyses do not allow clear conclusions to be reached. In essence, deletion of Med23 leads to an almost complete loss of functional HSCs as demonstrated in transplantation assays shown in Figure 4. Therefore, both functional and molecular analysis of phenotypic HSCs is irrelevant for HSC biology as these cells lack the HSC-defining property of self-renewal and likely no longer represent HSCs at all but rather some form of progenitor cell.

This is a legitimate concern. We performed single-cell RNA-seq analysis and found that Med23-deficient HSCs have high correlation with WT HSC ($R=0.96$) in Figure S8B. In addition, ATAC-seq analysis revealed that whole chromatin accessibility was comparable between WT HSCs and Med23-deficient HSCs (Figure S7F). These results indicate that although Med23-deficient HSCs lose self-renewal capacity in transplantation assays, Med23-deficient HSCs is not some form of progenitor cells.

To demonstrate lineage-bias it is essential to purify single-HSCs or MPPs pre-deletion of Med23 and then subject these cells to in vitro combined myeloid and lymphoid single-cell lineage assays with or without deletion. These experiments are challenging but without such single-cell analyses it is not possible to reach conclusions about lineage biases. Even then, it is crucial to exclude that the loss of lymphoid cells does not relate to a differentiation block in lymphoid development rather than a myeloid bias of multipotent cells. This again requires purification of HSCs pre-deletion of Med23, and then subjecting single cells to a lymphoid promoting culture and deleting at an early time point when full lymphoid commitment has occurred. If the hypothesis is correct then there should be no impact of Med23 loss in this assay.

We thank the reviewer's suggestion. We failed to get results from such single-cell experiment as it's technically challenging. Alternatively, we performed single-cell RNA-seq analysis instead to demonstrate lineage bias at single-cell level (Figure 5). In addition, we analyzed the lineage potential of HSC by population experiment through in vitro OP9 coculture system. Compared with WT HSCs with both myeloid and lymphoid potential, Med23-deficient HSCs only showed myeloid potential (Figure 2F, G).

Specific comments:

1. Med23 deficient mice develop thrombocytopenia, what is the explanation for this? Is Mk-priming of HSCs affected?

We performed GSEA and found Med23-deficient HSCs have Mk-priming state and the number of MK-primed progenitor cells (CD41⁺CD150⁺Lineage⁻cKit⁺Sca1⁻) is increased (Figure S2D and S7A). Considering that Med23 is essential for T cell function¹, we can't exclude the probability that Med23 also functions in platelet development and we have added an explanation in Page 6.

2. For quantification of B-cells data for CD19⁺ should be shown and for myeloid cells Mac1⁺/Gr1⁻ as well as Mac1⁺/Gr1⁺

We have analyzed B cells with CD19 marker according to the reviewer's suggestion. Results are similar to those done with B220 marker (Figure 2 below). For myeloid cells, we have measured the number of neutrophil cells (Gr1⁺Mac1⁺), immature myeloid cells (Gr1^{low}Mac1⁺) and other myeloid cells (Gr1⁻Mac1⁺) in bone marrows. As seen in Figure S2C, compared with WT, the number of neutrophil cells is decreased and the number of immature myeloid cells and the other myeloid cell are increased in Med23-deficient mice.

Figure 2. Absolute number of B cells (CD19⁺ cells) in spleens. At 21 days post administration of poly(I:C) B cells were quantified with CD19 marker in spleens, (n=5 p<0.001).

3. The expansion of CMP and GMP is at odds with the low normal numbers of myeloid cells in the blood. Are these CMPs and GMPs less effective at making mature myeloid cells?

As seen in Figure S2C, the development of neutrophil cells, immature myeloid cells and the other myeloid cell are somewhat disturbed in Med23-deficient mice,

suggesting that the loss of Med23 may affect myeloid cell development from myeloid progenitors, we have added an explanation in Page 6.

4. Same issue for the increased MEP population despite reduced platelet count and borderline reduced Hgb in these mice.

Considering that Med23 is a critical subunit of Mediator complex, Med23 may also function in the development of red blood cells from MEP, we have mentioned this in the revised manuscript (Page 6).

5. What is the total number of phenotypic LT-HSCs and at what timepoint following deletion are these analyses carried out? The analysis should be a number of weeks (minimum 6 weeks) after poly i:c administration.

The total number of phenotypic LT-HSC was shown in Figure 3A-D and Figure S3B.

We analyzed the phenotype around 21 days after poly I:C administration. We also analyzed the phenotype at 6 weeks and 9 weeks post poly I:C administration (Figure S3B).

6. The data in Fig4E-F appear to show ongoing chimerism when total BM cells and not HSCs are transplanted (unlike 4A and 4H). There is no loss of chimerism with time (unlike 4A and 4H). What is the explanation for this?

In Figure 4 E-G, the results were generated from mice transplanted with total bone marrow cells without competitor cells (CD45.1). In Figure 4A and 4H, the results were generated from mice transplanted with donor cells with competitor cells (CD45.1). The results indicate although Med23-deficient bone marrow cells have dramatically lost competitive capacity, they have short term self-renewal capacity. These results also suggest that Med23-deficient HSCs is not some form of progenitor cells.

7. The use of Mx1-Cre is highly problematic for the observed phenotype. It is quite possible that there is some form of combinatorial effect between poly i:c administration and loss of Med23. An alternative Cre line such as Vav-Cre or a different inducible system should also be used in parallel to exclude a poly i:c effect.

We thank the reviewer's suggestion which significantly improves our study. We used UBC-cre/ERT2 stain mouse to delete Med23. After tamoxifen administration we found the similar phenotype to those found from Mx1-cre/Med23 mice (Figure S4).

8. Myeloid (and platelet) biased HSCs expand with age. Mice should be aged and analysis carried out to study the impact of Med23 loss on the HSC aging phenotype.

We understand the reviewer's concern. Initially, we also thought that some phenotype of Med23-deficient mice mimics the phenotype of aged mice. We investigated the expression of Med23 in HSCs from young and old mice through real-time PCR and didn't find significant difference as shown in the below Figure 3A. Moreover, age-related signature gene set² is not significantly enriched in Med23-deficient HSCs through GSEA in the below Figure 3B.

Figure 3. Med23-deficient HSCs doesn't phenocopy aged HSCs. (A) Med23 expression in HSC of young (6-8weeks) mice and old (>18 months) mice. Normalized to *Gapdh*, N.S.: no significance. n=3. (B) GSEA showed that aging gene set was not significantly enriched in Med23-deficient HSCs.

9. What is the survival of primary mice following Med23 deletion? Do mice eventually develop BM failure. Is there any evidence of dysplasia in the myeloid cells?

Although Med23-deficient mice have reduced total number of bone marrow, the survival of primary mice is normal and the mice didn't develop total failure of BM following Med23 deletion, probably because that the native hematopoiesis, which rarely rely on the HSC population, is different from hematopoiesis under stresses or after transplantation.

Reference

1. Sun Y, Zhu XY, Chen XF, Liu HF, Xu Y, Chu YJ, et al. The mediator subunit Med23 contributes to controlling T-cell activation and prevents autoimmunity. *Nat Commun* 2014, 5: 5225.
2. Rossi DJ, Bryder D, Zahn JM, Ahlenius H, Sonu R, Wagers AJ, et al. Cell intrinsic alterations underlie hematopoietic stem cell aging. *P Natl Acad Sci USA* 2005, 102(26): 9194-9199.

Reviewer #3 :

Med23, a subunit of the transcriptional mediator complex, plays a role in controlling transcriptional profiles in different cell types. To investigate a potential role for Med23 in hematopoietic stem cells, an inducible mouse model is used to delete Med23 in the hematopoietic compartment. Based on their data the authors conclude that Med23 serves as a gatekeeper of the myeloid primed state of HSCs. The data presented with the mouse model are interesting, however based on these data, alternative explanations are possible as well. The initial observation in Figure 1 is that mice lacking Med23 in the hematopoietic system show reduced lymphoid cell numbers, but the number of myeloid cells in these mice stays the same. The authors conclude there is a myeloid bias in these mice and continue on this thought. Of course this decrease in lymphoid cells is changing the lymphoid-myeloid ratio in these mice. However, an alternative explanation of this phenotype could be that these mice have a compromised lymphoid compartment in stead of a myeloid bias, as has been shown for many immune-deficient mouse models. Throughout the manuscript the focus is on explaining the myeloid bias, ignoring any possible defect in lymphoid differentiation. In addition, the authors talk about a ‘primed state’ of HSCs. However the papers they are citing do not necessarily provide evidence for a ‘primed state’. Furthermore, the term ‘primed state’ is used for different HSC populations in different papers, it’s not a well-accepted term for a certain subpopulation of HSCs.

We understand the reviewer’s concern. Indeed, we can’t conclude that the Med23-deficient HSC under a “myeloid-primed state” simply based on the decrease number of lymphoid cells. In the manuscript, we found that 1) Med23-deficient mice have more myeloid progenitors (CMP and GMP) and less lymphoid progenitors, 2) Upon 5-FU treatment, med23-deficient mice show better survival than WT mice due to enhanced myeloid recovery, and 3) the Med23-deficient HSCs up-regulates the myeloid-related signature genes. We therefore conclude that Med23-deficient HSCs are myeloid-primed. In the revised manuscript, the new single cell analysis (Figure 5) further supports our conclusion that Med23-deficient HSCs express myeloid signature genes. Recently, a paper¹ published in *Nature* entitled “Clonal analysis of lineage fate in native hematopoiesis” suggested that a primed state exists in HSCs of native hematopoiesis by single cell sequencing. Together with other studies^{2,3,4}, we think that the primed state exists in HSC population and contribute to acute recovery.

Furthermore, there are several open questions/concerns:

- are all the analyses performed at the same timepoint after pIC injection (21 days post last injection)? why was this timepoint chosen? How does the phenotype develop over time? What about the survival of the mice? (Figure 1, 2, 3)

All the analyses were performed at 21 days post first injection of poly I:C except as otherwise described. We chose the time point “21 days post first injection” considering that the cell numbers of both Med23-deficient and WT HSCs reached maximum and cell cycle returned to quiescence around 21 days (Figure S3B and S3E). The survival of primary mice following Med23 deletion is normal although Med23-deficient mice have reduced total number of bone marrow.

- Have the authors confirmed the Med23 phenotype in a different inducible mouse model which is independent on pIC induced (and thus stress induction) deletion of Med23?

We thank the reviewer’s suggestion which significantly improves our study. We use UBC-cre/ERT2 stain mouse to delete Med23. After tamoxifen administration we found the similar results to those from Mx1-cre/Med23 mice (Figure S4).

- The panels in Figure 1 should be better labeled. It should be clear from the labeling of the y-axis which cell numbers are shown (myeloid, B cells, NK cells). Were the cell numbers in the peripheral blood also analyzed to get a better overview of overall cell numbers of differentiated cells in the mice?

We thank the reviewer’s suggestion and the y-axis has been labelled. The cell numbers of different mature cells in the peripheral blood have been shown in table 1.

- Figure 2D&E: did the authors check whether there might be a general increase in the CD34+ signal leading to a ‘change’ in populations? Why is there no absolute increase in myeloid cells if there is an increase in the number of myeloid progenitors?

We have analyzed the mean fluorescence intensity (MFI) of CD34 of GMP and CMP. There is no significant difference between WT and Med23-deficient GMP or CMP as shown in below Figure 4. One possibility to explain why there is no absolute increase in myeloid cells is that Med23 may also function in the development of myeloid cells from myeloid progenitors considering that Med23 is a critical subunit of Mediator complex. We have mentioned this in the revised manuscript (Page 6)

Figure 4. The Mean Fluorescence intensity (MFI) of CMP and GMP from WT and Med23-deficient mice. N.S.: no significance.

- There is quite some controversy in the field whether CD41 marks a subset of HSCs with strong myeloid potential. The level of CD41 in Figure 3A on wt HSCs is very high compared to published data, why?

The level of CD41 looks higher compared to published data probably due to the difference of mice age. The paper⁵ published in *blood* entitled “CD41 expression marks myeloid biased adult hematopoietic stem cells and increases with age” showed that the expression of CD41 increases with age. The mice were analyzed at age of at least 11 weeks, so the expression of CD41 is relative higher.

- Figure 3E would indicate a change in CD48 marker expression upon pIC treatment. Where do the CD50⁺CD48⁺ cells go in Med23KO mice? Whether the DN (CD150⁻CD48⁻) are the ST-HSCs is not very clear in the field. What happens to the expression of cKit and Sca-1, are there changes, both under homeostasis as well as upon pIC treatment?

Med23-deficient HSCs have impaired lymphoid potential in consistence with reduced number of CD48⁺CD150⁺LSKs which represent the lymphoid potential. So we think Med23-deficient HSCs have impaired differentiation into CD150⁺CD48⁺LSK. We have corrected our definition of ST-HSC (CD150⁻CD48⁻LSK) (Figure 3C). The expression of cKit and Sca-1 are slightly down-regulated after deletion of Med23 (Figure S3A and S4A).

- For the stress treatments in Figure 3 both the data under homeostasis as well as upon pIC treatment should be shown.

We are sorry for the misunderstanding because of our unclear description. In the manuscript, poly I:C injection was used for introduce of Med23 deletion but not the stress treatment. We have corrected our unclear description in the revised

manuscript (Page 6-7).

- Did the authors analyze cell cycle distribution with for example Ki67-Hoechst or shortterm BrdU incorporation (18h) as measurement for proliferation at baseline and upon pIC treatment? The BrdU data shown are longterm incorporation assays which are more of a stress readout.

We have performed short-term BrdU incorporation assay according to the reviewer's suggestion. Although long term incorporation assays showed that Med23-deficient HSCs proliferated faster than WT HSCs, short-time BrdU incorporation have no significant difference between WT and Med23-deficient HSCs (Figure S3E), which suggested that Med23-deficient HSCs undergo faster cell cycle upon stresses such as 5-day chase of BrdU, while stay quiescent under homeostasis.

- Are Med23 KO HSCs cycling more and could this be part of the reason why they perform worse in a transplant assay (Figure 4A)? Did they try to transplant HSCs i.f.? Did the authors perform limiting dilution experiments to analyze the difference in HSC frequency?

According to the literature⁶, when quiescence is disrupted, HSCs might display premature exhaustion, impaired self-renewal. Under homeostasis, both Med23-deficient HSCs and WT HSCs are quiescent (Figure S3E), and the myeloid-primed state of Med23-deficient HSCs is reason why they lose competition when we transplanted them (Figure 4A-D). We have performed limiting dilution experiments. The results indicate that Med23-deficient mice have less frequency of HSCs than WT HSCs (Figure S6B).

- In Figure 4H-K the authors start pIC injection only 4 weeks post transplantation. At this time point there is no good recovery yet in the transplanted mice and reconstitution is not fully established yet. What happens when they would wait until 12 weeks post transplant and then inject with pIC?

We have repeated this kind of experiments and started poly I:C injection at 12 weeks post transplantation according to review's requirement. Similar results were found (Figure S6A).

- In Figure 4L-M the percentage of cells are shown, not the total nr of cells. The alternative explanation here could again be that the number of B and T cells are reduced due to reduced lymphoid cell production and thus the % of myeloid cells are higher.

We understand the reviewer's alternative explanation. The percentage change is mainly used as a hint, and many other results support our conclusion as

mentioned above.

- The data analysis of the RNAseq in Figure 5 are very limited and rather one sided. How about a lymphoid signature? Is the myeloid signature the only major difference? In the text they talk about Uhrf1-deficient FL-HSPCs????

We have done more analysis, such as GSEA analysis, ATAC-seq analysis and single-cell RNA seq analysis to strength our study. The single-cell RNA seq analysis clearly showed a relatively more enrichment of myeloid specific genes in Med23-deficient HSCs, with much lesser enrichment of CLP- and lymphoid-specific genes (Figure 5 B).

We thank the reviewer's nice point, the mistake "Uhrf1-deficient FL-HSPCs" has been corrected (Page 10).

- What is expression of Med23 in the different hematopoietic cell populations under homeostasis in stead of upon 5FU (Figure 6A)?

We have measured the expression of Med23 in different hematopoietic cells including HSCs, and as seen in Figure S1A, Med23 is widely expressed among hematopoietic cells.

- Figure S6 is missing several panels: hard to say something about this data without the actual data present.

We have corrected the mistake (Figure S9).

- In Figure 6 the different populations are analyzed upon 5FU treatment. However 5FU is causing a lot of marker changes, which are still present 4 and 7 days post 5FU. Therefore it's difficult to judge these data. The raw FACS plots would be very useful here.

We have added the raw FACS plots in Figure S9A according to the reviewer's suggestion.

- The suggestion based on the data shown in Figure 6H is pure speculative, there is no data to back this up.

Upon myeloablative stresses, such as 5-FU treatment, HSCs enter into myeloid-primed state having phenotype of higher CD41 expression (Figure 6H) through down-regulation of expression of Med23 (Figure 6A). Med23-deficient HSCs (CD41⁺HSCs) are myeloid-primed and improve recovery after 5-FU treatment.

- In the discussion the authors speculate about what happens to Med23 expression

upon recovery from stress and how Med23 deficient HSCs behave during the recovery phase. These are easy hypothesis to test in their model and would increase the message of their story.

We thank the reviewer's suggestion. It is difficult to define the recovery time point and recovery phase since the decreased Med23 expression is detected at day 5 after 5-FU treatment, we have alternatively performed more analysis to strength our conclusion that Med23-deficient HSCs are myeloid-primed, such as single cell RNA-seq analysis, ATAC-seq analysis, and *in vitro* differentiation analysis with OP9 system (Figure 2F and 2G, Figure 5, and Figure S7F and S7G).

- How would Med23 function as a gatekeeper other than the general explanation of it control transcription? A bit more mechanism here would be informative.

We thank the reviewer's suggestion. We have discussed more in the Discussion part (Page 14-15).

- the text would benefit a lot from an English grammar correction. Some of the sentences are difficult to read because of problems with the English.

The English has been improved accordingly.

Reference

1. Rodriguez-Fraticelli AE, et al. Clonal analysis of lineage fate in native haematopoiesis. *Nature* 553, 212 (2018).
2. Miyamoto T, et al. Myeloid or lymphoid promiscuity as a critical step in hematopoietic lineage commitment. *Dev Cell* 3, 137-147 (2002).
3. Mansson R, et al. Molecular evidence for hierarchical transcriptional lineage priming in fetal and adult stem cells and multipotent progenitors. *Immunity* 26, 407-419 (2007).
4. Hu M, et al. Multilineage gene expression precedes commitment in the hemopoietic system. *Genes & development* 11, 774-785 (1997).
5. Gekas C, Graf T. CD41 expression marks myeloid-biased adult hematopoietic stem cells and increases with age. *Blood* 121, 4463-4472 (2013).
6. Cheshier SP, Morrison SJ, Liao XS, Weissman IL. In vivo proliferation and cell cycle kinetics of long-term self-renewing hematopoietic stem cells. *P Natl Acad Sci USA* 96, 3120-3125 (1999).

Reviewers' comments:

Reviewer #1 (Remarks to the Author):

The authors have made an impressive effort revising the manuscript, addressing comments and adding new data. I believe that the work is ready for publication.

Reviewer #2 (Remarks to the Author):

Whilst the authors have addressed some of the more minor concerns raised, my fundamental concern with this work remains the same. The data are entirely consistent with loss of Med23 causing a major loss of functional (not phenotypic) HSCs and expansion of myeloid progenitors which offer some short term protection following 5FU treatment. The single cell RNA-seq data and functional data are entirely consistent with this. It also remains unclear whether the reduction in lymphoid cells is caused by a differentiation block or an alteration in priming of multipotent cells.

The data presented are interesting and the phenotype is quite dramatic but the interpretation of the data remains flawed.

Reviewer #3 (Remarks to the Author):

The manuscript has greatly improved by the additional new data and changes made in the text. The extra data using the UBC Cre mouse system have strengthened the data, showing it is not linked to any pIC induced side effect. The single cell analyses have strengthened the myeloid bias point the authors would like to make, even though there could still be a reduced lymphoid differentiation component explaining part of the phenotype as well. This has not been completely ruled out.

There are still some minor points:

- until which age have the authors followed the mice? What happens in aged Med23 deficient mice? Does the myeloid bias become more profound?
- The CD41 argumentation related to increase of CD41 with age does not really relate to their data. CD41 expression on HSCs is indeed increased in old mice, but their mice are 11 weeks old, that's not an old mouse, that's a young adult.
- Limiting dilution shows reduced functional Med23KO HSCs; this is probably the main reason why they perform worse in a competitive transplantation, not the myeloid-primed state of the Med23KO HSCs as they authors keep arguing.

Reviewer #1 :

The authors have made an impressive effort revising the manuscript, addressing comments and adding new data. I believe that the work is ready for publication.

We thank the reviewer for his/her positive comments and encouragement.

Reviewer #2 :

Whilst the authors have addressed some of the more minor concerns raised, my fundamental concern with this work remains the same. The data are entirely consistent with loss of Med23 causing a major loss of functional (not phenotypic) HSCs and expansion of myeloid progenitors which offer some short term protection following 5FU treatment. The single cell RNA-seq data and functional data are entirely consistent with this. It also remains unclear whether the reduction in lymphoid cells is caused by a differentiation block or an alteration in priming of multipotent cells.

The data presented are interesting and the phenotype is quite dramatic but the interpretation of the data remains flawed.

We thank the reviewer for his/her interest in our study.

Self-renewal and differentiation are closely correlated to each other in HSCs. Under physiological conditions, HSCs precisely balance the self-renewal capacity and differentiation potential through orchestrating the transcriptional networks^{1,2}. In this study, we highlight that Med23 is one of the key transcriptional regulators that balance the self-renewal capacity versus myeloid differentiation potential through controlling the myeloid-primed state of HSCs. Phenotypically, Med23-deficient HSCs harbor enhanced myeloid differentiation potential (Figure 2, 3, 6) with impaired self-renewal capacity (Figure 4). Mechanistically, the “phenotypic” Med23-deficient HSCs (CD150⁺CD34⁺CD48⁻LSKs) are correlated with WT HSCs (Supplementary Fig. 7b,f) but with low expression of self-renewal related genes and premature expression of myeloid signature genes (Figure 5). Together with the observation of HSCs’ behavior and protection following 5FU treatment (Figure 6), we conclude that Med23-deficient HSCs are in myeloid-primed state. The two major phenotypes could be connected closely by this “primed state” just like two sides of one coin. If those two phenotypes are separated, Med23-deficient HSCs should only show the specific transcriptome of WT HSCs without myeloid signatures, or only show the myeloid specific genes without similar signatures of HSCs. We have added more discussion in the Discussion Section (Page 16).

Regarding to the reviewer’s second concern, we have checked the number of MPP4 which is the known earliest lymphoid progenitors that derived from the HSCs and MPPs. Notably, Med23-deficient mice show strongly reduced MPP4 population (Supplementary Fig. 2e, in revised manuscript) in the bone marrow compared with WT mice, which is consistent with the phenotype of decreased common lymphoid progenitors (CLPs) in Med23-deficient mice. These results indicate that the reduction in lymphoid cells could be caused by alteration in priming of multipotent cells though we can’t rule out other differentiation blocks

from lymphoid progenitors (CLPs) to mature lymphoid cells. We have added the MPP4 data in the Results Section (Page 6 in the revised manuscript).

Reference

1. Novershtern N, Subramanian A, Lawton LN, Mak RH, Haining WN, McConkey ME, et al. Densely Interconnected Transcriptional Circuits Control Cell States in Human Hematopoiesis. *Cell* 2011, 144(2): 296-309.
2. Gottgens B. Regulatory network control of blood stem cells. *Blood* 2015, 125(17): 2614-2620.

Reviewer #3 :

The manuscript has greatly improved by the additional new data and changes made in the text. The extra data using the UBC Cre mouse system have strengthened the data, showing it is not linked to any pIC induced side effect. The single cell analyses have strengthened the myeloid bias point the authors would like to make, even though there could still be a reduced lymphoid differentiation component explaining part of the phenotype as well. This has not been completely ruled out.

We thank the reviewer for his/her positive comments.

The lineage commitment of HSCs is balanced and enforced differentiation of HSCs into one lineage often dampens the potentials of other lineages^{1, 2, 3, 4}. Med23-deficient HSCs have premature expression of myeloid specific genes with the reduced expression of CLP genes or lymphoid-primed genes (Figure 5b), suggesting that the lymphoid potential of Med23-deficient HSCs was inhibited. To further investigate the impaired lymphoid differentiation, we have checked the number of MPP4, which is the known earliest lymphoid progenitors that derived from the HSCs and MPPs. Notably, dramatically reduced MPP4 population was found (Supplementary Fig. 2e, in the revised manuscript) in the bone marrow of Med23-deficient mice compared with WT mice, which is consistent with the phenotype of decreased common lymphoid progenitors (CLPs) in Med23-deficient mice. These results indicate that the reduction in lymphoid cells could be caused by alteration in priming of multipotent cells though we can't rule out other differentiation block from lymphoid progenitors (CLPs) to mature lymphoid cells. We have added the MPP4 data in the Results Section (Page 6 in the revised manuscript).

There are still some minor points:

- until which age have the authors followed the mice? What happens in aged Med23 deficient mice? Does the myeloid bias become more profound?

We analyzed the phenotype of several relatively old mice (about 1 year old). Med23-deficient mice showed dramatically reduced bone marrow cellularity and HSC number (below Figure 1A and B) compared with WT mice. Moreover, the biased myeloid differentiation potential of Med23-deficient HSCs was more profound in the 1-year-old mice compared with mice at 11-week old (below Figure 1C).

- The CD41 argumentation related to increase of CD41 with age does not really relate to their data. CD41 expression on HSCs is indeed increased in old mice, but their mice are 11 weeks old, that's not an old mouse, that's a young adult.

We are sorry for our unclear statement. We wanted to emphasize that the CD41 expression becomes higher with age in the last version. To get a better understanding of the CD41 expression level, we analyzed the CD41 expression in HSCs with the help of isotype controls (below Figure 2B). Our data is similar to a published study done by Gekas et al⁵. (below Figure 2A), which showed that a large proportion (around 90%) of HSCs (CD150^{high}) express CD41 even at 8 weeks. And Med23-deficient HSCs (CD150^{high}) showed higher CD41 expression level compared with WT HSCs (Figure 3e and 3f). In the study, we distinguished CD41⁺ HSCs from CD41⁻ HSCs by using the CD41 expression level in CD150^{neg} LSKs as negative control, which is exactly what Gekas et al. did in their study (below Figure 2C). Using this gating strategy, we obtained similar results as we showed in Figure 3e.

Figure 1. Myeloid potential of Med23-deficient HSCs from one-year-old mice. (A and B) Absolute number of total bone marrows (A) and HSCs (B) from mice at 1-year post poly I:C injection (n=3). (C) The myeloid and lymphoid potential of mice at 3 weeks or 1-year post poly I:C injection. Normalized to myeloid/lymphoid potential of WT mice at 3 weeks post poly I:C injection (n=3). The data are means \pm S.D., for all panels: *P < 0.05; **P < 0.01 by Student's *t*-test.

- Limiting dilution shows reduced functional Med23KO HSCs; this is probably the main reason why they perform worse in a competitive transplantation, not the myeloid-primed state of the Med23KO HSCs as they authors keep arguing.

The “phenotypic” HSCs (CD150⁺CD34⁻CD48⁻ LSKs) are increased at day 21 after Med23 deletion, which indicates that the reconstitution failure result from the impaired function rather than the reduced number of Med23-deficient HSCs. In addition, the “phenotypic” Med23-deficient HSCs are correlated with WT HSCs (Supplementary Fig. 7f,8b) but with low expression of self-renewal related genes and premature expression of myeloid signature genes (Figure 5). The myeloid-primed transcriptional program initiated in Med23-deficient HSCs enforces them to enter an irreversible differentiation process, leading to impaired self-renewal capacity. Our findings are consistent with previous researches

showing forced differentiation by perturbing the transcriptional networks can lead to impaired self-renewal capacity^{6,7,8}.

Figure 2. Flow cytometry analysis of CD41 expression.. [redacted] (B and C) from our experiments, 46.1% CD41⁺CD150⁺ LSK was found within the CD150⁺ LSK population (C).

Reference

1. Collombet S, van Oevelen C, Ortega JLS, Abou-Jaoude W, Di Stefano B, Thomas-Chollier M, *et al.* Logical modeling of lymphoid and myeloid cell specification and transdifferentiation. *Proceedings of the National Academy of Sciences of the United States of America* 2017, **114**(23): 5792-5799.
2. Iwasaki H, Akashi K. Myeloid lineage commitment from the hematopoietic stem cell. *Immunity* 2007, **26**(6): 726-740.
3. Iwasaki H, Mizuno S, Wells RA, Cantor AB, Watanabe S, Akashi K. GATA-1 converts lymphoid and myelomonocytic progenitors into the megakaryocyte/erythrocyte lineages. *Immunity* 2003, **19**(3): 451-462.
4. Stehling-Sun S, Dade J, Nutt SL, DeKoter RP, Camargo FD. Regulation of lymphoid versus myeloid fate 'choice' by the transcription factor Mef2c. *Nat Immunol* 2009, **10**(3): 289-296.
5. Gekas C, Graf T. CD41 expression marks myeloid-biased adult hematopoietic stem cells and increases with age. *Blood* 2013, **121**(22): 4463-4472.
6. Chan WI, Hannah RL, Dawson MA, Pridans C, Foster D, Joshi A, *et al.* The Transcriptional Coactivator Cbp Regulates Self-Renewal and Differentiation in Adult Hematopoietic Stem Cells. *Mol Cell Biol* 2011, **31**(24): 5046-5060.
7. Sun L. Id2 Is Required for the Self-Renewal and Proliferation of Hematopoietic Stem Cells. *Blood* 2015, **126**(23).

8. Gottgens B. Regulatory network control of blood stem cells. *Blood* 2015, **125**(17):2614-2620.

REVIEWERS' COMMENTS:

Reviewer #2 (Remarks to the Author):

The authors have not addressed my key concerns. The manuscript is highly dependent on a phenotypic definition of HSC but in the disease model; it is not correct to assume that phenotypically defined HSCs are still functional HSCs. In fact, all data shown are consistent with the phenotypically defined Med23 deficient HSC population actually being dominated by contaminating myeloid progenitor cells and not containing significant numbers of functional HSC. The phenotypic HSC cell population in Med23 deficient mice does not possess defining properties of HSC i.e. Med23 deficient phenotypic HSC lack self renewal capability and multipotency.

Experiments with reversible knockdown of Med23 would definitively deal with this question as when Med23 expression is restored, normal HSC function should resume according to the authors conclusions. The more likely possibility is that most/all functional HSCs are lost on Med23KO and restoration of Med23 expression will fail to restore normal hematopoiesis.

Alternatively, the authors could deal with this by a rewrite of their manuscript, with careful and full acknowledgement of different interpretations of their data, including change in the title, without the need for any new data.

Reviewer #3 (Remarks to the Author):

The authors have addressed the remaining concerns.

Reviewer #2

The authors have not addressed my key concerns. The manuscript is highly dependent on a phenotypic definition of HSC but in the disease model; it is not correct to assume that phenotypically defined HSCs are still functional HSCs. In fact, all data shown are consistent with the phenotypically defined Med23 deficient HSC population actually being dominated by contaminating myeloid progenitor cells and not containing significant numbers of functional HSC. The phenotypic HSC cell population in Med23 deficient mice does not possess defining properties of HSC i.e. Med23 deficient phenotypic HSC lack self renewal capability and multipotency.

Experiments with reversible knockdown of Med23 would definitively deal with this question as when Med23 expression is restored, normal HSC function should resume according to the authors conclusions. The more likely possibility is that most/all functional HSCs are lost on Med23KO and restoration of Med23 expression will fail to restore normal hematopoiesis.

Alternatively, the authors could deal with this by a rewrite of their manuscript, with careful and full acknowledgement of different interpretations of their data, including change in the title, without the need for any new data.

According to the reviewer's requirement, we have toned down our conclusion in the revised manuscript and changed the title into "Med23 serves as a gatekeeper of the myeloid potential of hematopoietic stem cells". In addition, we have reorganized our manuscript and discussed the phenotypic Med23-deficient HSCs accordingly (Page 15).

Reviewer #3 (Remarks to the Author):

The authors have addressed the remaining concerns.

We thank the reviewer for his/her positive comments